

# First evaluation of the GEMS formaldehyde retrieval algorithm against TROPOMI and ground-based column measurements during the in-orbit test period

Gitaek T. Lee[1], Rokjin J. Park[1], Hyeong-Ahn Kwon[2], Eunjo S. Ha[1], Sieun D. Lee[1], Seunga Shin[1], Myoung-Hwan Ahn[3], Mina Kang[3], Yong-Sang Choi[3], Gyuyeon Kim[3], Dong-Won Lee[4], Deok-Rae Kim[4], Hyunkee Hong[4], Bavo Langerock[5], Corinne Vigouroux[5], Christophe Lerot[5*], Francois Hendrick[5], Gaia Pinardi[5], Isabelle De Smedt[5], Michel Van Roozendael[5], Pucai Wang[6], Heesung Chong[7], Yeseul Cho[8], and Jhoon Kim[8]

[1]School of Earth and Environmental Science, Seoul National University, Seoul, Republic of Korea
[2]Department of Environmental & Energy Engineering, University of Suwon, Suwon, Republic of Korea
[3]Department of Climate and Energy Systems Engineering, Ewha Womans University, Seoul, Republic of Korea
[4]Envionment Satellite Center, National Institute of Environmental Research, Incheon, Republic of Korea
[5]Royal Belgian Institute for Space Aeronomy (BIRA-IASB), Brussels, Belgium
[6]Institute of Atmospheric Physics, Chinese Academy of Sciences (CAS), Beijing, China
[7]Atomic and Molecular Physics Division, Harvard-Smithsonian Center for Astrophysics, Cambridge, Massachusetts, USA
[8]Department of Atmospheric Sciences, Yonsei University, Seoul, Republic of Korea
*Now at Constellr, Brussels, Belgium

*Correspondence to*: Rokjin J. Park (rjpark@snu.ac.kr) and Hyeong-Ahn Kwon (hakwon@suwon.ac.kr)

**Abstract.** The Geostationary Environment Monitoring Spectrometer (GEMS) onboard GEO-KOMPSAT 2B was successfully launched in February 2020 and has monitored Asia. We present the first evaluation of the operational GEMS formaldehyde (HCHO) vertical column densities (VCDs) during the in-orbit test period (IOT) (August–October 2020) and onward by comparing them with the products from Tropospheric Monitoring Instrument (TROPOMI), Fourier-Transform Infrared (FTIR), and Multi-Axis Differential Optical Absorption Spectroscopy (MAX-DOAS) instruments. During the in-orbit

test period, the GEMS HCHO VCDs reproduced the observed spatial pattern of TROPOMI VCDs over the whole domain (r=0.62) with high biases (10–16 %). In the afternoon, GEMS VCDs were too high over the west side of the tropics. We corrected this issue by adding polarization sensitivity vectors of the GEMS instrument as an additional fitting parameter in the retrieval algorithm. Using observed radiances from clear-sky pixels as the reference spectrum in the spectral fitting



significantly contributed to reducing artifacts in radiance references, resulting in 10–40 % lower HCHO VCDs over the latitude including cloudy areas in the updated GEMS product. We find that the agreement between the two is much higher in Northeast Asia (r=0.90), including the Korean peninsula and East China. GEMS HCHO VCDs well captured the seasonal variation of HCHO mainly driven by biogenic emissions and photochemical activities but showed larger variations than those of TROPOMI over coastal regions (Kuala Lumpur, Singapore, Shanghai, and Busan). In addition, GEMS HCHO VCDs showed consistent hourly variations with MAX-DOAS (r=0.79) and FTIR (r=0.85) but were lower by 30–40 % relative to the ground-based observations. Different vertical sensitivities between GEMS and ground-based instruments caused these systematic biases. The use of averaging kernel smoothing method reduces the low biases by about 10 to 15 % (NMB: -48.5 % to -32.4 %, -39.1 % to -27.3 % for MAX-DOAS and FTIR, respectively). The remaining discrepancies are due to multiple factors, including spatial colocation and different instrumental sensitivities, which need further investigation using inter-comparable datasets.

## 1. Introduction

Non-methane volatile organic compounds (NMVOCs) are precursors of surface ozone ($O_3$), a harmful pollutant, that affects the human respiratory system (Shrubsole et al., 2019) and plant photosynthesis activities (Matyssek and Sandermann, 2003). NMVOCs also play critical roles in the formation of secondary organic aerosols (DiGangi et al., 2012). They are emitted from anthropogenic and biogenic sources (Vrekoussis et al., 2010). The latter is more important globally but has significant uncertainty (Abbot et al., 2003; Palmer et al., 2001). Previous studies (Cao et al., 2018; Choi et al., 2022; Palmer et al., 2003) have attempted to reduce this uncertainty using observational constraints, including satellite-derived vertical column densities (VCDs) of formaldehyde (HCHO), which is produced by the oxidation of NMVOCs and used as a proxy of NMVOCs.

Starting with the Global Ozone Monitoring Experiment (GOME) launched in 1995 (Chance et al., 2000), HCHO has been observed globally by sun-synchronous low earth orbit (LEO) satellites. Scanning Imaging Absorption Spectrometer for Atmospheric Chartography launched in 2002 had measured HCHO VCDs with 60 km × 30 km spatial resolutions at nadir for 2002–2012 (Wittrock et al., 2006). Observations from these satellites provided global and regional distributions of HCHO VCDs and were effectively used to constrain NMVOCs emissions in biogenic source dominant regions worldwide (Stavrakou et al., 2009). However, the spatial resolutions of those satellites were too coarse to detect local pollution plumes.



Subsequent LEO satellites, including the Ozone Monitoring Instrument (OMI), Tropospheric Monitoring Instrument (TROPOMI), Global Ozone Monitoring Experiment 2A (GOME-2A), and Ozone Mapping and Profiler Suite (OMPS) nadir mapper, have adopted much finer spatial resolutions (5.5 km × 3.5 km ~ 80 km × 40 km) to observe local pollution plumes and can be effectively used to provide observational constraints for biogenic and anthropogenic sources globally (Veefkind et al., 2012; De Smedt et al., 2015, 2021; Li et al., 2015; González Abad et al., 2016; Levelt et al., 2018; Nowlan et al., 2023;
Kwon et al., 2023). Moreover, De Smedt et al. (2015) examined diurnal characteristics of global HCHO VCDs from GOME-2 and OMI with different overpass times (GOME-2: 9:30, OMI: 13:30, local time), showing that afternoon HCHO VCDs are higher than the morning over most regions with exceptions in the tropical rainforest. However, limited by the overpass time, these LEO satellites provide observations at most once a day, which could be significantly compromised by the cloud, especially in East Asia.

In East Asia, anthropogenic emissions of NMVOCs are also highly uncertain, causing significant errors in chemical transport models (Park et al., 2021). Cao et al. (2018) used satellite HCHO products from GOME-2A and OMI with an inverse modeling technique to estimate "top-down" anthropogenic VOCs (AVOCs) emissions in China. Recently, Choi et al. (2022) showed a large underestimate of VOCs emissions (29–115 %) in the anthropogenic emission inventory in East Asia using the top-down inversion with OMI and OMPS HCHO VCDs. Kwon et al. (2021) estimated top-down AVOCs emissions using
aircraft HCHO vertical column observations during the Korea-US cooperative air quality campaign and demonstrated the efficacy of remote sensing HCHO VCDs observations to estimate AVOC emissions in polluted urban areas. However, the previous studies based on LEO satellite or aircraft observation products had not considered the continuous daytime variability of HCHO VCDs to the emission estimates, suggesting the necessity of deploying a geostationary satellite over East Asia.

        The Geostationary Environment Monitoring Spectrometer (GEMS), launched on 19 February 2020 by the Korean
Ministry of Environment, has started hourly observations of trace gases (NO$_2$, SO$_2$, O$_3$, HCHO, CHOCHO) and aerosols with 3.5 km × 8 km pixels or co-added pixels over Seoul, Korea (Kim et al., 2020). Kwon et al. (2019) developed the GEMS HCHO retrieval algorithm and evaluated its performance with OMI Level 1B data before the launch. In this study, we describe several updates implemented in the operational GEMS HCHO retrieval algorithm and present its evaluation results by comparing GEMS HCHO VCDs with TROPOMI products and ground-based observations, including Multi-Axis Differential Optical
Absorption Spectroscopy (MAX-DOAS) and Fourier-Transform Infrared (FTIR), during the in-orbit test (IOT) period (August–October 2020) and onward. We also performed sensitivity tests of GEMS HCHO to input parameters to improve the retrieved column's precision.



## 2. Operational GEMS HCHO algorithm description

GEMS is located at 128.25° E and conducts hourly observations for Asia (5° S–45° N, 75–145° E). The spectral range
of GEMS covers 300–500 nm with a spectral resolution of 0.6 nm and a wavelength interval of 0.2 nm. GEMS's spatial
resolution is 3.5 km × 8 km for $NO_2$, $O_3$, HCHO, and aerosols at Seoul, and relatively weak absorbers, including sulfur dioxide
($SO_2$) and glyoxal (CHOCHO), use 2 × 2 or 4 × 4 co-added pixels (7 km × 16 km and 14 km × 32 km, respectively) to increase
the signal-to-noise ratio. HCHO as a weak absorber in the UV spectral region can be retrieved at 2 × 2 or 4 × 4 co-added pixels.
To reduce representation error and discern heterogeneous plumes, however, HCHO is retrieved in the original spatial resolution
(Souri et al., 2023; Kwon et al., 2023). Detailed information on the GEMS instrument can be found in Kim et al. (2020).

Kwon et al. (2019) described the GEMS HCHO retrieval algorithm (version 0.3), consisting of three-step processes:
pre-processing, spectral fitting, and post-processing. The pre-processing includes calibration of radiances and irradiances from
Level 1C data and convolution of absorption cross-sections. The spectral fitting derives slant column densities (SCDs) using
the basic optical absorption spectroscopy algorithm (Chance, 1998), a non-linearized fitting method, to solve the Lambert–
Beer's equation. Finally, the post-processing performs background corrections by model column from unpolluted clean sector
and conversion from SCD to VCD by applying air mass factor (AMF) (Palmer et al., 2001). Here, we briefly describe several
updates in the retrieval algorithm compared to Kwon et al. (2019).

We updated the absorption cross-sections and a fitting window for GEMS HCHO retrieval shown in Fig. 1 and Table
1, based on our sensitivity tests discussed in Sect. 3. The operational retrieval uses the fitting window of 329.3–358.6 nm,
which is within the range of 328.5–346.0 nm (De Smedt et al., 2008, 2012, 2015) and 328.5–359.0 nm (De Smedt et al., 2018)
used for GOME-2 and TROPOMI. Variables of GEMS HCHO Level 2 product are detailed in Table S1.

In the spectral fitting, the measured radiances over clean regions, referred to as a radiance reference, can be used instead
of the solar irradiance. Using the radiance reference can minimize ozone and bromine monoxide interferences in the
stratosphere (Kwon et al., 2019). Radiance references sampled from the Pacific Ocean were used for the LEO satellites (De
Smedt et al., 2008, 2021; González Abad et al., 2015, 2016). In the case of GEMS, a radiance reference is computed by
averaging measured radiances from the clean pixels, mainly consisting of ocean pixels, from the easternmost part within its
domain as a function of cross-tracks (north-south direction) and is used as the default option for the spectral fitting with an
alternative option using irradiance references. We discuss the sensitivity of the retrieval to regions sampled for radiance
references and compare the operational products with those using irradiance references in Sect. 3.



In the case of using radiance references, we need to add HCHO background concentrations included in measured radiances, called a background correction. As described in Kwon et al. (2019), the background correction is to add slant columns simulated by a chemical transport model over the reference sector to retrieved slant columns as a function of latitude. For GEMS HCHO, GEOS-Chem simulation in 0.25° × 0.3125° spatial resolutions provides zonal mean HCHO VCDs over the radiance reference sector. Next, model HCHO VCDs are converted to SCDs by multiplying with pre-calculated zonal mean

AMFs.

        During the IOT, GEMS scanning plans were changed from a fixed scan area to varying scan areas from east to west to obtain more observations in western areas, including India. Therefore, GEMS cannot sufficiently obtain clean pixels from the Pacific Ocean every hour. We widened the sampling regions for the radiance reference from 143–150° E (Kwon et al., 2019) to 120–150° E (Fig. S1). The new reference sector partially includes polluted areas over East China, Korea, and Japan and

could affect the background contributions to VCD. We examined the impact of the changed reference sector on the background correction in Sect. 4.

        High reflectance conditions, such as thick clouds, can affect the radiance references. We select clear-sky pixels with minimal clouds (cloud radiance fraction < 0.4). Figure 2 compares the radiance references with (Fig. 2a) and without (Fig. 2b) cloud masking. Radiance references sampled in the minimal cloud condition show fewer artifacts with exceptionally high

radiances (~230 W cm$^{-2}$ cm$^{-1}$ sr$^{-1}$) along latitudes and wavelengths, implying the better quality of the reference spectra. However, despite its extended reference sector in operation, GEMS HCHO has often failed to reserve sufficient clean radiance pixels from the reference sector under cloudy conditions, resulting in a few missing tracks. We use mean radiance references from the previous two days' observations to fill the missing latitudinal points of the reference spectra. As shown in Fig. 2c, running mean radiance references efficiently recover the missing tracks, presenting consistent spatial and spectral distributions to Fig.

2a. Consequently, the use of the updated radiance references to HCHO spectral fitting reduces high VCDs of the previous retrieval by 10–40 % over the entire scan domain, showing negative normalized mean bias (NMB) (-22.9 %).

        Figure 3 shows the fitted optical depth as a function of wavelengths for a specific pixel in Midwest China at 12:45 Korean Standard Time (KST) on 3 August 2020. The black solid line shows the fitted optical depth of HCHO, and the red solid line represents optical depth together with fitting residuals. In the case of uncertainty of the fitted slant columns, we

estimate the uncertainty due to random noise in operational Level 1C radiances. Averaged random uncertainty and its fitting root-mean-square (RMS) of GEMS radiances on 3 August 2020 are 7.7 × 10$^{15}$ molecules cm$^{-2}$ and 9.7× 10$^{-4}$, which are comparable with the synthetic radiances (Random uncertainty: 9.1 × 10$^{15}$ molecules cm$^{-2}$; RMS: 1.2 × 10$^{-3}$) and OMI Level



1B data (Random uncertainty: $1.1 \times 10^{16}$ molecules cm$^{-2}$; RMS: $1.2 \times 10^{-3}$) computed by Kwon et al. (2019). The random uncertainty on the GEMS HCHO retrieval is also consistent with TROPOMI ($6.0 \times 10^{15}$ molecules cm$^{-2}$) (De Smedt et al., 140    2021)

Kwon et al. (2019) employed a look-up table approach to efficiently calculate AMFs to convert fitted SCDs to VCDs. The table consists of pre-calculated scattering weights, based on monthly mean a priori profiles simulated from GEOS-Chem with a spatial resolution of 2° × 2.5°. We updated it using monthly mean hourly profiles simulated from GEOS-Chem with a much finer spatial resolution of 0.25° × 0.3125° and the most up-to-date anthropogenic and biomass burning emission 145    inventories in Asia (Table 2). We compare the two discrete AMFs derived from the initial and operational algorithms to evaluate the impacts of the updated a priori profiles (Fig. 4a and b, respectively). Their absolute differences in Fig. 4c present decreased AMFs over Southeast Asian megacities, which increase their VCDs, and the opposite behaviors over the ocean pixels above 5° N. Disparities between the two AMFs are mainly due to the updates of anthropogenic emission inventories pertaining to metropolitan cities and biomass burning occurrences over East Asia. In addition, the fine spatial resolution of the 150    new a priori profile better separates ocean pixels in AMF calculation, resulting in high AMFs and eventually low VCDs over the coastal areas such as Borneo and Hanoi.

## 3. Sensitivity tests

This section conducts several sensitivity tests of GEMS HCHO retrievals to key input parameters, including polarization correction, fitting window, and irradiance reference. Unlike TROPOMI or OMI, GEMS is not equipped with a 155    polarization scrambler. Observed radiances can be sensitive to polarization, especially for the wavelengths of HCHO absorption (Choi et al., 2021; Kotchenova et al., 2006; Choi et al., 2023). Therefore, polarization correction needs to be applied in the retrieval algorithm. We include the instrument's polarization sensitivity vectors as a pseudo absorption cross-section in the spectral fitting. The polarization sensitivity values shown in Fig. 5 were measured before the launch of the GEMS instrument and provided a single spectrum for the central part of the charge-coupled-device (Choi et al., 2023).

Figure 6 shows monthly mean hourly GEMS HCHO VCDs with and without polarization correction during IOT. High VCDs ($> 1.5 \times 10^{16}$ molecules cm$^{-2}$) occur over the west of the tropics without polarization correction, especially in the late afternoon (15:45 KST) (Fig. 6b). After applying the polarization correction, these high values are eliminated, as shown in Fig. 6a, with slightly increased columns over Northeast Asia. In addition, the relative differences (Fig. 6c) induced by polarization



correction result in about 30 % variations in HCHO VCDs from the scan domain's east to the west edge side. These spatial
patterns could occur due to the geometric dependency of polarization vectors, as previously presented by Choi et al. (2021).
Polarization correction could also affect the fitting window sensitivity to the retrieved slant columns because it is not linearly
considered in the spectral fitting. We evaluate the fitting window of GEMS HCHO in the next step to discern the algorithm's
retrieval sensitivity under polarization correction.

To find an optimized fitting window, we conduct a sensitivity test of the retrieved HCHO SCDs with polarization
correction by varying lower limit of the fitting window from 327 to 329.5 nm and the upper limit from 354 to 360 nm with the
wavelength interval of 0.2 nm over the reference sector (120–150° E). As shown in Fig. 7a, negative values of the mean SCDs
over the reference sector are shown over the entire fitting window except for upper limits at 358.5–359.5 nm. Based on the
low RMS of the fitting residuals and fitting uncertainty shown in Figures 7b and 7c, we chose the fitting window of 329.3–
358.6 nm for the GEMS HCHO operational retrieval.

We conduct a sensitivity test of the GEMS HCHO retrieval using solar irradiances as a reference spectrum. The use of
irradiance for trace gas retrieval often causes stripe patterns, due to the cross-track dependent factors including diffuser, dark
current, noise, and other factors along the scan tracks for the satellite (Chan Miller et al., 2014). We perform a de-striping
process (Lerot et al., 2021), subtracting the median values of each cross-track with background correction. Figure 8 compares
the GEMS HCHO VCDs retrieved using irradiance and radiance references during IOT. HCHO VCDs using irradiance spectra
have good agreement (r=0.87) with those using radiance references but are 20–50 % higher VCDs in the high latitudes (> 40°
N). Also, HCHO products using the irradiance reference show 10–50 % higher fitting RMS (~ $2.5 \times 10^{-3}$) and random
uncertainties (~ $8 \times 10^{15}$ molecules cm$^{-2}$) than those using the radiance reference. Therefore, results using a radiance reference
are almost identical with those using an irradiance with the fitting parameter, and a radiance reference is used as a reference
spectrum in the operational retrieval. However, as discussed in Section 4.1, the reference sector including polluted regions can
lead to small contribution of the retrieved slant columns to total column in some regions. It is required to investigate the
possibility to use an irradiance and to update the reference sector minimizing inclusion of polluted regions in a future study.



## 4. GEMS HCHO VCDs evaluation

### 4.1. Comparison with TROPOMI

190        In this section, we compare the GEMS HCHO VCDs with those from TROPOMI, which have a similar spatial resolution (5.5 km × 3.5 km). We filter out unqualified values of TROPOMI HCHO VCDs using the "Quality Assurance (QA)" variable (QA < 0.5). For GEMS, we use operational Level 2 HCHO product (version 2.0) and select pixels in a "good" quality flag (FinalAlgorithmFlags = 0) with cloud radiance fraction less than 0.4 and less effect from SZA (< 70°) and VZA (< 70°) for validation. GEMS pixels are collocated with TROPOMI at the overpass time of 13:30 local time (LT). Then, GEMS and

TROPOMI data are re-gridded using an area-weighted mean with a spatial resolution of 0.1° × 0.1° to create a comparable dataset.

        Figure 9 shows GEMS HCHO VCDs against TROPOMI during the IOT. HCHO VCDs over the continent are high during summer due to active photochemistry and high biogenic VOC emissions. Large anthropogenic emissions also contribute to high VCDs in megacities (e.g., Shanghai, Beijing, Hong Kong, and Seoul). These characteristics are well captured by GEMS

observations, which are consistent with TROPOMI with a correlation coefficient of 0.62 over the entire domain. Over north-eastern Asia, GEMS HCHO VCDs have better agreement with TROPOMI (r=0.90).

        However, GEMS VCDs are lower by $4 \times 10^{15}$ molecules cm$^{-2}$ than TROPOMI over the north-western edge of the scan domain with high viewing zenith angles (VZA > 60°), as shown in Fig. 9c. When we use GEMS pixels under low viewing zenith angles (VZA < 60°), the correlation coefficient between GEMS and TROPOMI increases from 0.62 to 0.66, showing

increased NMB (17 %➔ 22 %) by the eliminated low biases from the north-western edge. The low GEMS VCDs could be attributed to the longer light path with high VZA, which is more susceptible to light scattering, making spectral fitting more uncertain. In addition, GEMS is a geostationary satellite sensor and has the sensitivity of retrievals with respect to SZA. The value of cloud fraction increases exponentially for SZA values above 40° and has large uncertainty above 60° (Kim et al., 2021). Multiple scatterings by gases and aerosols with a longer light path could also affect AMF calculations. Further

investigations are required to consider scattering effects on the SCDs and AMFs for high geometric conditions.

        The changes in the GEMS scan domain also affect the construction of radiance reference and eventually the retrieved HCHO VCDs. In October 2020, GEMS changed its afternoon scan schedule for 12:45–13:45 KST from the nominal (100–147° E) to the full west (FW) region (77–133° E), and the available sector for the reference spectrum was narrowed down. We examine the GEMS HCHO VCDs retrieved using the radiance references sampled from the FW scan to assess the impact of a



narrower reference sector. GEMS HCHO VCDs derived from FW radiance references during IOT entirely show 5–20 % lower values than the operational GEMS HCHO in Fig. 9 and present decreased NMBs (17 % → 13 %) against TROPOMI, showing enhanced negative biases over Midwest China. This comparison implies that further investigations should be conducted for reserving sufficient radiance reference pixels to prevent potential biases in the spectral fitting results.

      GEMS cloud properties are available at 331, 360, and 420 nm as well as 477 nm. GEMS and TROPOMI use observed

radiances at different wavelength bands to derive cloud properties ($O_4$ at 477 nm for GEMS vs. $O_2$–A band at 760 nm for TROPOMI), retrieving different physical meanings of cloud fractions and cloud pressures (Kim et al., 2023; Loyola et al., 2018). This discrepancy makes different results in the AMF calculation, despite being observed at the same time. This study utilized a cloud radiance fraction of 331 nm as cloud fraction, which is the nearest to the HCHO fitting window. To exclude the cloud dependency on HCHO AMF in the comparison between GEMS and TROPOMI, we define cloud-free VCDs ($VCDs_{cf}$)

by applying AMFs under a cloud-free assumption, which was introduced by De Smedt et al. (2021). Figures S2d and S2h illustrate $VCDs_{cf}$, displaying similar agreements to those from comparisons in Fig. 9 but with slight changes in statistics. In particular, the presence of the cloud mainly affects Southeast Asian cities and less polluted mid-latitude areas with 4–8 % lower NMBs compared to the original VCDs. It is probably due to cloudy conditions related to the Asian rainy monsoon from August to October affecting AMF calculation.

230       Figure 10 compares the seasonal variations of monthly mean GEMS and TROPOMI HCHO VCDs in 22 cities shown in Fig. S3, which have high population densities, petrochemical complexes, or power plants in East Asia. We use averaged values over pixels within a 20 km × 20 km grid box centering on each city's center. Panels on the first and second rows in Fig. 10 present Southeast Asian cities, and those on the third and fourth rows are Northeast Asian cities. GEMS shows good agreements with TROPOMI with correlation coefficients of r=0.58–0.82.

235       In Southeast Asian cities (Vientiane, Ho Chi Minh, Hanoi, Bangkok, Yangon, and Phnom Penh), the highest HCHO VCDs occur in spring due to biomass burning. In other cities, HCHO VCDs peak in summer, resulting from high photochemical reactivity with increased biogenic VOCs emissions. GEMS captures well this seasonal variation. GEMS HCHO VCDs in Tokyo show a relatively poor correlation coefficient (r=0.58) with TROPOMI because of the insufficient GEMS pixels from the westward scan domain from May 2021. For this reason, Tokyo shows only a correlation coefficient from

August 2020–April 2021. The total numbers of sampled pixels over Japan (Tokyo: 60, Osaka: 76) are almost one-third of the overall mean pixel count for the entire cities (mean pixel number: 200.4).



For VCDs$_{cf}$, shown in Fig. S4, monthly mean GEMS and TROPOMI HCHO columns in Southeast Asian cities such as Ho Chi Minh, Hanoi, Taipei, and Kuala Lumpur increased by $5 \times 10^{15}$ molecules cm$^{-2}$ from February to March. On the other hand, VCDs$_{cf}$ over Northeast Asian cities do not show remarkable changes in their concentrations. These are very similar to the results of De Smedt et al. (2021), who reported that cloud-free assumption could highly reduce existing biases in comparing satellites over South Asian regions and less effectively works in mid-latitude polluted areas. VCDs$_{cf}$ from GEMS and TROPOMI present higher correlation coefficients (r=0.6–0.85) for all cities than VCDs including some cloudy conditions, showing more distinctive seasonal and annual variations as the clear sky assumption excludes the cloud dependency on the vertical columns.

Background correction also plays a crucial role in the computation of VCDs. However, obtaining clean radiance references from uncontaminated background pixels for GEMS is challenging in Northeast Asia. This difficulty arises because the GEMS scan domain predominantly covers the continental area at high latitudes resulting in higher background columns. We evaluate the effect of background contributions from GEMS HCHO a priori profile using VCDs without background correction (VCD$_0$). Figure S5 shows the same comparison between GEMS and TROPOMI except for VCD$_0$. In Southeast Asia, TROPOMI shows 10–15 %p higher contributions of VCD$_0$ to VCDs than GEMS, showing consistent correlation coefficients of r=0.51–0.73. However, in Northeast Asia, the difference in VCD$_0$ contributions between TROPOMI and GEMS has widened by 70 %p with lower correlation coefficients of r=0.36–0.7. Consequently, the simulated background model values significantly contribute to the final VCD columns in the retrieval in Northeast Asia.

## 4.2. Direct and harmonized comparison with ground-based MAX-DOAS and FTIR observations

We evaluate GEMS HCHO VCDs by comparing them with ground-based MAX-DOAS and FTIR observations at Xianghe (116.96° E, 39.75° N) in China. We use GEMS observations averaged in pixels within a 20 km × 20 km grid box centering on the ground observatory, following De Smedt et al. (2021) who determined the similar size radius circle as an optimal value in TROPOMI and MAX-DOAS HCHO comparison. To make inter-comparable datasets among GEMS and ground observations, we apply a smoothing method (Rodgers and Connor, 2003) and a priori substitution following equations 2 and 3 of Vigouroux et al. (2020) using averaging kernel and a priori profile of GEMS.

Figure 11a presents daily and monthly mean HCHO VCDs for GEMS and MAX-DOAS during GEMS observation time (08:45–15:45 KST). GEMS shows a good correlation (r=0.79) of daily mean VCDs with MAX-DOAS but also presents



low NMB (-48.5 %) in the direct comparison without any corrections. Averaging kernel smoothing with a priori profile

correction reduces the differences between GEMS and MAX-DOAS. As shown in the orange line in Fig. 11a, the negative

NMB between GEMS and MAX-DOAS decreases (NMB=-48.5 % → -32.37 %), and the linear regression slope becomes

close to one (slope=0.5 → 0.76) with a consistently high correlation coefficient (r=0.79 → 0.8), which is consistent with the

results in De Smedt et al. (2021), indicating the different vertical sensitivities between two remote-sensed products. Therefore,

the difference in the instrument's vertical sensitivity should be considered when comparing two remote-sensed products. For

example, Fig. S8 shows the averaging kernels for GEMS, MAX-DOAS, and FTIR over Xianghe, and MAX-DOAS is more

sensitive near the surface than FTIR and GEMS.

Figure 11b shows that GEMS HCHO VCDs have good agreement with those from FTIR with high correlation

coefficients (r=0.85) in the direct comparison. NMB between GEMS and FTIR is -39.09 %, which is less negative than those

of MAX-DOAS. While the correlation coefficient between smoothed FTIR and GEMS VCDs is slightly lower than that from

the direct comparison (r=0.85 → 0.82), NMB (-39.09 % →-27.25 %) and RMSE ($6 \times 10^{15}$ → $5.72 \times 10^{15}$) are reduced.

Although the vertical sensitivity of FTIR is already similar to satellite observations (De Smedt et al., 2021), the above results

show that the effect of averaging kernel smoothing is still not negligible.

The remaining discrepancies between GEMS and the two ground-based observations become maximum during the

summertime, possibly due to the dilution of HCHO in the large GEMS area. The HCHO production from isoprene oxidation

in summertime can be localized, inducing a steep spatial gradient. GEMS pixel observing the MAX-DOAS station covers a

much larger area, leading to diluted HCHO VCDs, especially when the observing area has a high HCHO concentration.

Figure 12 shows diurnal variations of GEMS and MAX-DOAS HCHO VCDs. De Smedt et al. (2015) showed the

diurnal variation of HCHO from MAX-DOAS at Xianghe from 2010 to 2013, with two peaks occurring in the morning (06–

08 LT) and afternoon (14–16 LT) due to the anthropogenic emissions in rush hour and the high insolation with increasing

290    temperature, respectively. The diurnal variation of VCDs from GEMS is consistent with the previous result from De Smedt et

al. (2015), with the increasing trend of HCHO VCDs in the daytime.

Figure S6 shows the same analysis from FTIR, which presents consistent diurnal variation with GEMS. Smoothed

FTIR VCDs are $2.5 \times 10^{15}$ molecules $cm^{-2}$ lower than the original VCDs throughout the daytime. FTIR shows decreasing

HCHO VCDs from 14 LT while those from MAX-DOAS continuously increase. The discrepancy between MAX-DOAS and

295    FTIR appears because the FTIR products have 2–10 times fewer observation numbers than MAX-DOAS, especially in the





afternoon (14–16 LT). In Fig. S7, MAX-DOAS HCHO VCDs sampled in the FTIR observation time show consistent diurnal variations with those from FTIR.

## 5. Conclusions

The first geostationary satellite observation of HCHO was started by GEMS, which enables investigating the diurnal variability of HCHO over East Asia. In this study, we improved the initial GEMS HCHO retrieval algorithm and evaluated its performance in operation. The initial algorithm caused high positive biases in the slant columns from the spectral fitting, mainly led by radiance references constructed under cloudy conditions with high reflectance. We removed the existing artifacts of the sampled radiance references by collecting clear-sky pixels over the reference sector. In addition, GEMS also showed high positive biases over the western tropics nearby the Bay of Bengal and Indonesia under high solar zenith angles. These high biases are primarily due to the interference of polarized lights from aerosols and gases. We considered the polarization sensitivity vectors of the GEMS instrument, which is not equipped with a polarization scrambler, as a pseudo-absorber in the spectral fitting and reduced the high biases of HCHO VCDs in the afternoon. Based on these modifications, we performed a sensitivity test for the GEMS fitting window and concluded that 329.3–358.6 nm is an optimized range to fit the slant column.

We evaluated the GEMS HCHO using the TROPOMI product. During IOT, GEMS and TROPOMI showed a good agreement (r=0.65) of HCHO VCDs over the entire scan domain, with especially higher correlation coefficients in East Asia (r=0.9). However, we found that the changes in the reference sector highly affect the retrieved columns' precision. We tested three-day running mean radiance references to reduce missing tracks in observations, which provided a better quality of the sampled spectra. Although the irradiance references can also be utilized as a reference spectrum, as mentioned in Sect. 3, they showed much higher fitting RMS and random uncertainty than the radiance references. To use solar irradiances as a reference spectrum, we need to study an efficient way to correct the retrieved slant columns from the irradiance references.

We found a good representation of seasonality and regional characteristics of GEMS HCHO among the major cities, showing active emissions from biogenic and anthropogenic sources over East Asia. Using VCDs under the cloud-free assumption, we determined that the effect of the cloud products in AMF calculations does not significantly contribute to the retrieval quality of polluted Northeast Asian cities, similar to the results from De Smedt et al. (2021). However, there are also high variations in differences between GEMS and TROPOMI over coastal areas such as Kuala Lumpur, Singapore, Shanghai, and Busan. These can be associated with the scene heterogeneity problem of measured radiances caused by the heterogeneous



terrain heights or materials, such as mountains or coastal areas over the scanning track. This problem is detectable in a satellite product with a fine spatial resolution because the large pixel size dilutes an error from the problematic area (De Smedt et al., 2021). Richter et al. (2018) presented a correction method by considering the heterogeneity factor in the spectral fitting, showing better performance of OMI $NO_2$ VCDs. Further studies need to be conducted to characterize the effects of the heterogeneity factor on the GEMS observation.

GEMS HCHO VCDs were also consistent with ground-based MAX-DOAS and FTIR observations in Xianghe. GEMS produce approximately 30 % lower columns than MAX-DOAS but show high correlations and good seasonality during a year. We harmonized MAX-DOAS and FTIR products using GEMS a priori profile and averaging kernel. MAX-DOAS and FTIR recalculated VCDs showed evident declines with a better correlation coefficient against GEMS after harmonization. We found that using an identical a priori profile with vertical smoothing enables a precise intercomparison, partially resolving the systematic discrepancy between satellite and ground-based instruments.

In addition, a representation error, a mismatch between the high value of the point measurements and satellite pixels under polluted areas, could be one of the possible causes for the low values of the satellite-retrieved columns (Brasseur and Jacob, 2017). Ground-based observation products over background regions should be jointly compared with the GEMS HCHO in examining the sensitivity of GEMS pixels to point measurement by pollution level of the target regions. Recently, Souri et al. (2022) presented an effective way to deal with the spatial heterogeneity between satellite and ground-based observations by using kriging interpolation which statistically converts point data to gridded values. They reduced systematic biases of $NO_2$ VCDs between OMI and ground-based Pandora observations, but this method needs at least three ground observation points nearby the satellite pixel to be applied. More ground-based observations must be conducted to tackle the underlying limitations in satellite validation.



**Data availability.**

The GEMS Level 1C data are available on request from the National Institute of Environmental Research (NIER) – Environmental Satellite Center (ESC). The GEMS Level 2 products are available at https://nesc.nier.go.kr/ko/html/index.do (last access: 22 August 2023). The TROPOMI HCHO product is available at https://disc.gsfc.nasa.gov/datasets/ (last access: coi22 August 2023) (De Smedt et al., 2021). MAX-DOAS HCHO and FTIR HCHO products are available at https://www-air.larc.nasa.gov/missions/ndacc/data.html?RapidDelivery=rd-list (last access: 22 August 2023) (Vigouroux et al., 2020).


**Author contributions.**

GTL, RJP, and HAK designed the study, carried out the analyses, and wrote the manuscript. ESH, SDL, and SS participated in the algorithm development. MHA and MK provided the GEMS Level 1B product. JK, HC, and YC provided the GEMS Level 1C product. YSC and GK provided the GEMS cloud product. DWL, DRK, and HH supported the GEMS instrument

management. IDS and CL provided the TROPOMI HCHO product. MVL, FH, GP carried out the MAX-DOAS measurement at Xianghe. BL, CV, and PW carried out the FTIR measurement at Xianghe.

**Competing interests.**

Michel Van Roozendael is an editor of the journal.


**Special issue statement.**

This article is part of the special issue "GEMS: first year in operation (AMT/ACP inter-journal SI)". It is not associated with a conference.

**Acknowledgements.**

The authors thank the GEMS science team and the Environment Satellite Center (ESC) of National Institute of Environmental Research (NIER) for supporting the GEMS HCHO retrieval algorithm development.

**Financial support.**



This research was supported by a grant from the Nation Institute of Environmental Research (NIER), funded by the Korea
      Ministry of Environment (MOE) of the Republic of Korea (NIER-2023-04-02-050).

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






**Table 1. Summary of operational GEMS algorithm attribute, parameters for radiance fitting, and parameters for the air mass factor (AMF) lookup table which is following (Kwon et al., 2019) except for absorption cross-section, fitting window, and reference spectrum sector.**

| | | |
|---|---|---|
| GEMS system attributes | Spectral range | 300–500 nm |
| | Spectral resolution | < 0.6 nm |
| | Wavelength sampling | < 0.2 nm |
| | Signal-to-noise ratio | > 720 at 320 nm > 1500 at 430 nm |
| | Field of regard (FOR) | ≥ 5000 (N/S) km × 5000 (E/W) km (5° S–45° N, 75–145° E) |
| | Spatial resolution (at Seoul) | < 3.5 km × 8km for gas and aerosol |
| | Duty cycle | ~ 8 times per day |
| | Imaging time | ≤ 30 min |
| Radiance fitting parameters | Fitting window (calibration window) | 329.3–358.6 nm (326.3–361.0 nm) |
| | Reference | Three-day mean measured radiances from easternmost swaths (120–150° E) under clear-sky condition (cloud radiance fraction < 0.4) |
| | Solar reference spectrum | Chance and Kurucz (2010) |
| | Absorption cross-sections | HCHO at 300 K (Chance and Orphal, 2011) $O_3$ at 223 K and 243 K (Serdyuchenko et al., 2014) $NO_2$ at 220 K (Vandaele et al., 1998) BrO at 228 K (Wilmouth et al., 1999) |



| | | |
|---|---|---|
| | | $O_4$ at 293 K (Finkenzeller and Volkamer, 2022) |
| | Ring effect | Chance and Kurucz (2010) |
| | Common mode | Online common mode from easternmost swaths (120–150° E) for a day |
| | Scaling and baseline polynomials | Third order |
| AMF lookup table parameters | Longitude (degree) (n=33) | 70 to 150 with 2.5 grid |
| | Latitude (degree) (n=30) | -4 to 54 with 2.0 grid |
| | Solar zenith angle (degree) (n=9) | 0, 10, 20, 30, 40, 50, 60, 70, 80 |
| | Viewing zenith angle (degree) (n=9) | 0, 10, 20, 30, 40, 50, 60, 70, 80 |
| | Relative azimuth angle (degree) (n=3) | 0, 90, 180 |
| | Cloud top pressure (hPa) (n=7) | 900, 800, 700, 600, 500, 300, 100 |
| | Surface albedo (n=7) | 0, 0.1, 0.2, 0.3, 0.4, 0.6, 0.8, 1.0 |

**Table 2. Summary of the input options of a priori profiles for the GEMS HCHO algorithm.**

| Version | Initial | Operational |
|---|---|---|
| Model | GEOS-Chem (v9-01-02) (Bey et al., 2001) | GEOS-Chem (v13) (Bey et al., 2001) |
| Period | 2014 | August 2020–July 2021 |
| Horizontal resolution | 2° × 2.5° | 0.25° × 0.3125° |
| Vertical layers | 47 | 47 |



| Meteorology | Modern-Era Retrospective Analysis for Research and Applications (MERRA) (Rienecker et al., 2011) | GEOS-FP (Goddard Earth Observing System -Forward Processing) assimilated meteorology |
|---|---|---|
| Emission inventory | **Biogenic**<br>- Model of Emissions of Gases and Aerosols from Nature (MEGAN) version 2.1 (Guenther et al., 2006)<br>**Anthropogenic**<br>- Database for Global Atmospheric Research (EDGAR) version 2.0 inventory (Olivier et al., 1996)<br>- Mosaic fashion with the Intercontinental Chemical Transport Experiment Phase B (INTEX-B) (Zhang et al., 2009)<br>**Monthly biomass burning**<br>Global Fire Emissions Database (GFED) version 3 inventory (van der Werf et al., 2010) | **Biogenic**<br>- Model of Emissions of Gases and Aerosols from Nature (MEGAN) version 2.1 (Guenther et al., 2006)<br>**Anthropogenic**<br>- Community Emissions Data System (CEDS) v2018-04 (Hoesly et al., 2018)<br>- KORUS version 5 over Asia (Woo et al., 2020)<br>**Monthly biomass burning**<br>Global Fire Emissions Database (GFED) version 4 inventory (Giglio et al., 2013) |





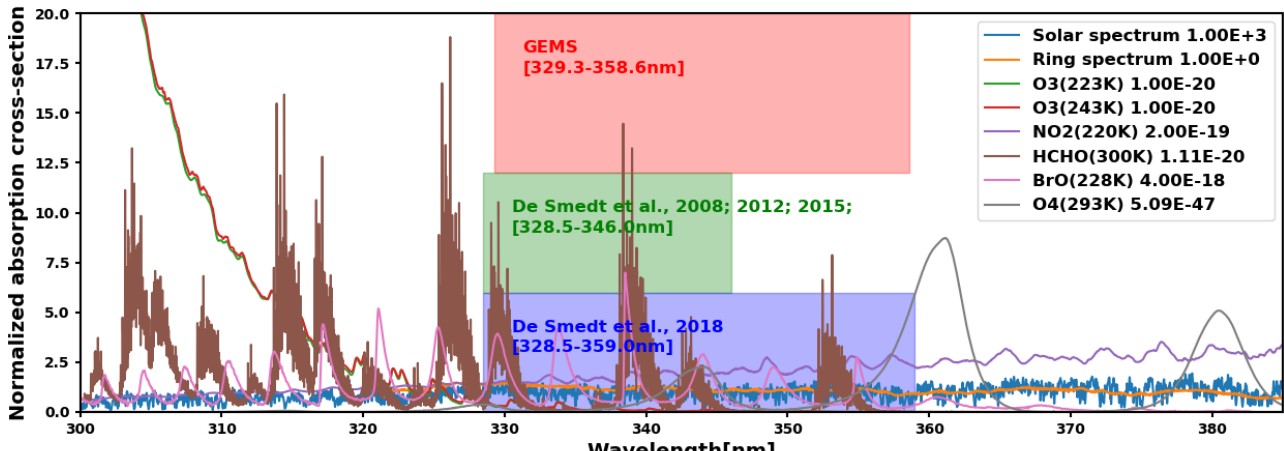

**Fig. 1. HCHO Fitting windows of the GEMS and TROPOMI from previous studies (De Smedt et al., 2008, 2015, 2012, 2018). Solid lines are reference absorption sections used in the GEMS HCHO algorithm. Values next to the legend represent normalization factors.**





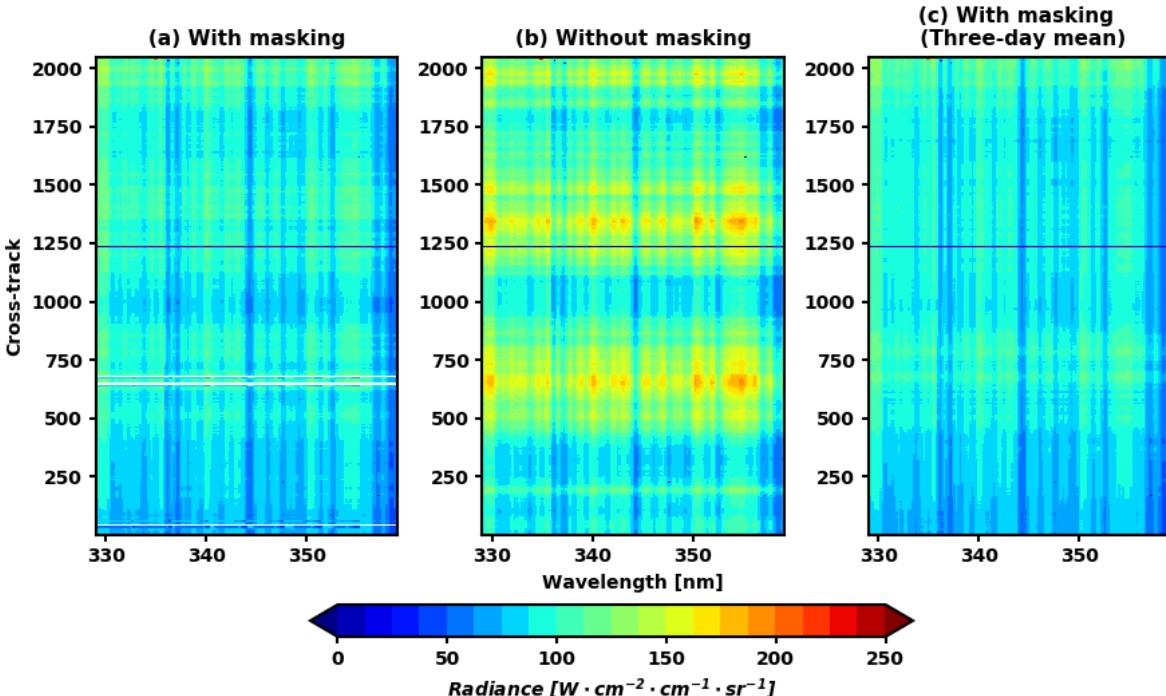


**Fig. 2. Latitudinally averaged radiance references of GEMS (03:45 UTC (12:45 KST), 6 December 2020): With cloud masking (cloud radiance fraction > 0.4) (a), without cloud masking (b), and cloud masking with three-day mean radiances (c). Shadings are radiance spectra. Radiance spectra in 1233–1241 cross tracks have bad L1C quality flags.**




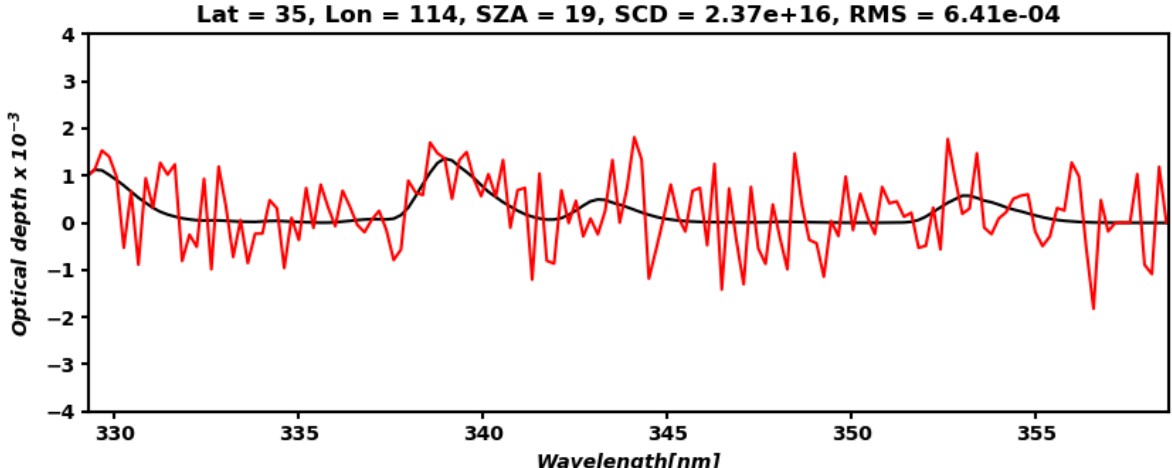

**Fig. 3 Fitted HCHO optical depth (black solid line) and optical depth plus the fitting residuals (red solid line) of the operational GEMS HCHO retrieval algorithm in Midwest China at 12:45 KST (03:45 UTC), 3 August 2020.**

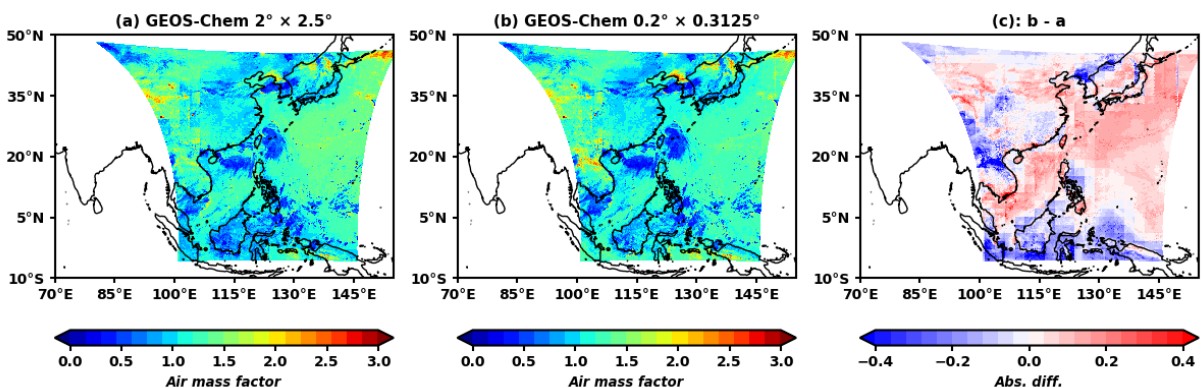


**Fig. 4. GEMS HCHO Air mass factor (3 August 2020, 12:45 KST (03:45 UTC)): The GEMS algorithm with initial a priori profile (a) , GEMS with updated a priori profile (b), and absolute differences of b - a (c).**



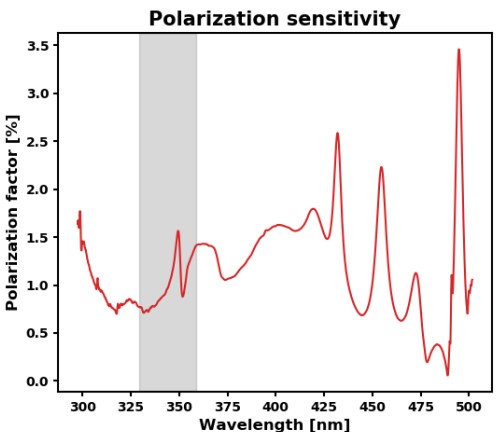

**Fig. 5. Polarization sensitivity vector of GEMS instrument (shaded area: fitting window of the GEMS HCHO).**

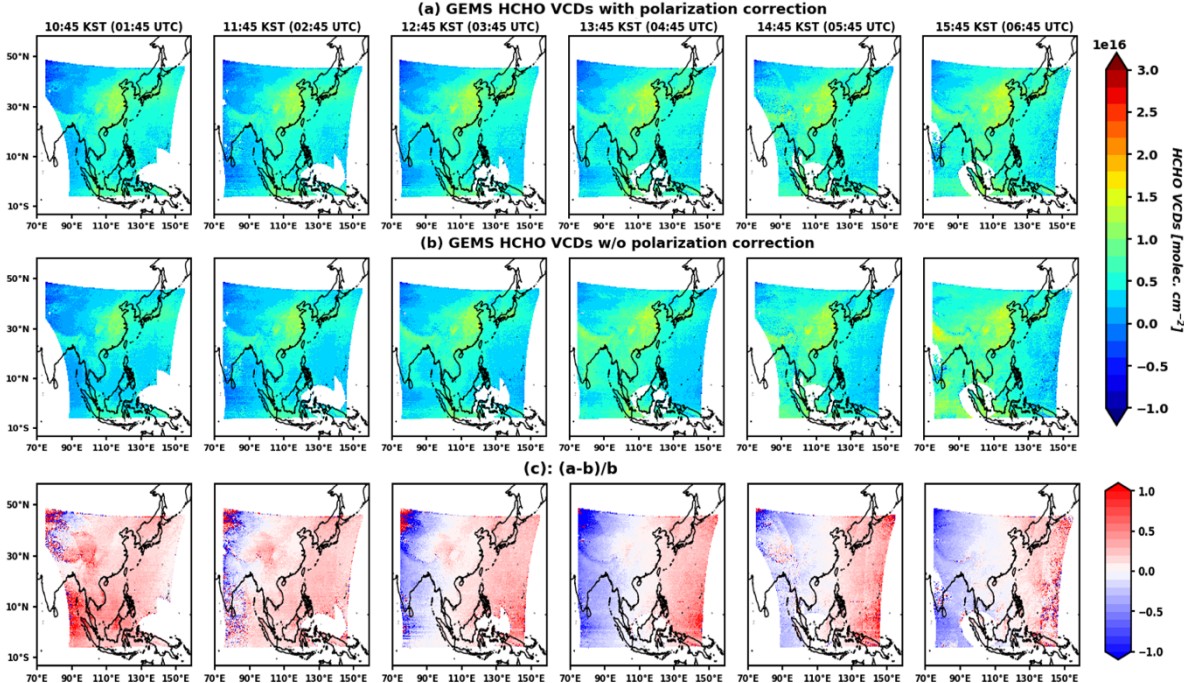





**Fig. 6. Average time dependence of GEMS HCHO VCDs with (a) and without (b) polarization correction, and their relative differences (c) ((a-b)/b) during IOT.**


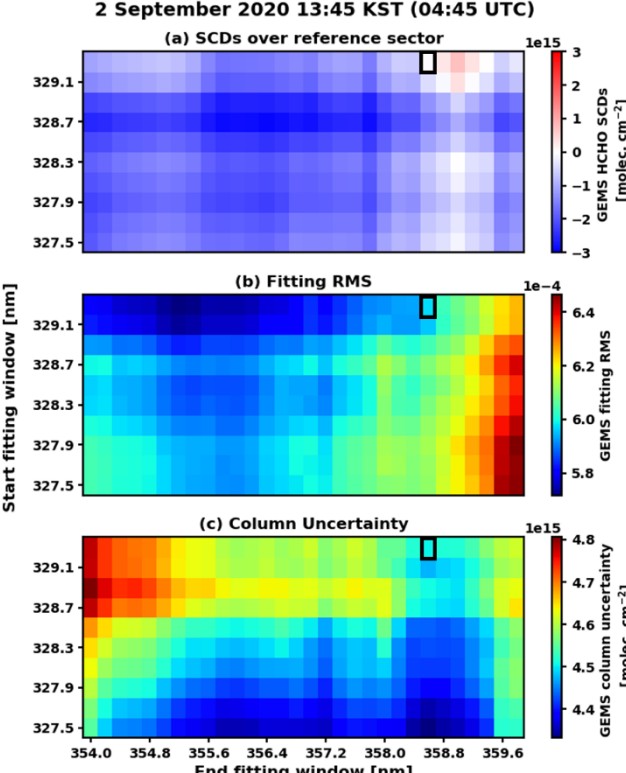

**Fig. 7. HCHO SCDs over reference sector (120–150° E) (a), fitting RMS (b), and column uncertainty (c) retrieved from GEMS for 2 September 2020. All pixels satisfy the clear-sky condition (cloud radiance fraction < 0.4) and the "good" main data quality flag. Background correction is not applied in this result. The optimum fitting window is shown in the black solid rectangle.**




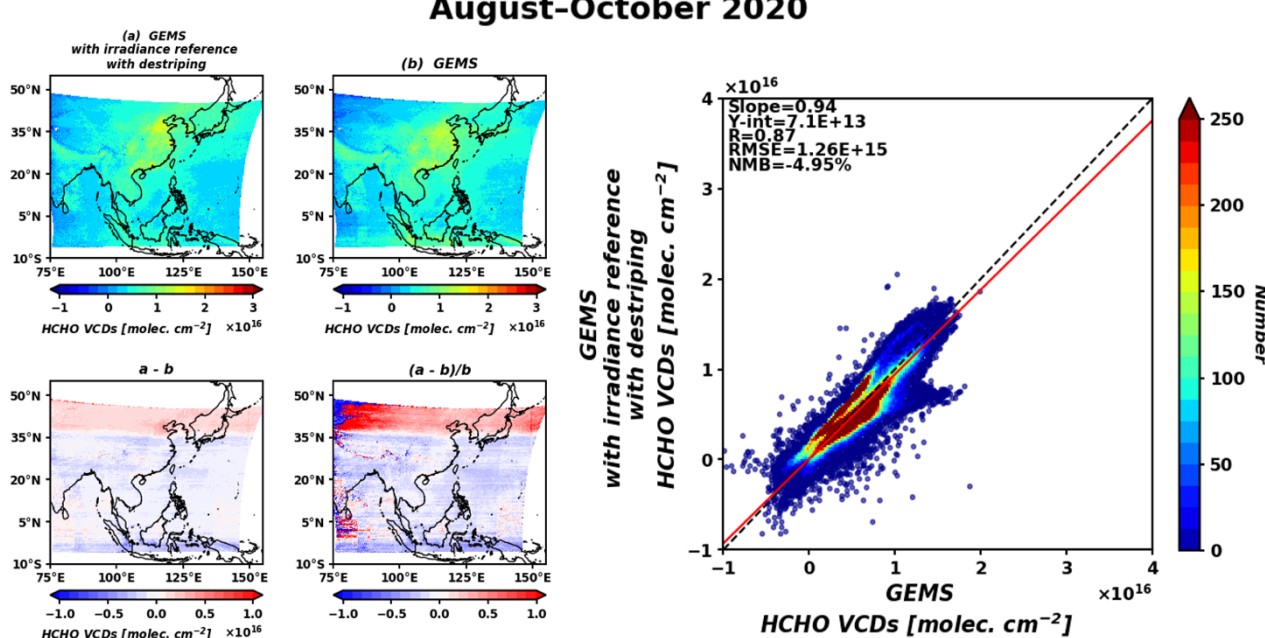

**Fig. 8. Mean HCHO VCDs from the GEMS using measured irradiance with de-striping (upper left), radiance references (upper right) for 09:45–15:45 KST (00:45–06:45 UTC) during IOT (August–October 2020), and scatter plots (right) between them.**



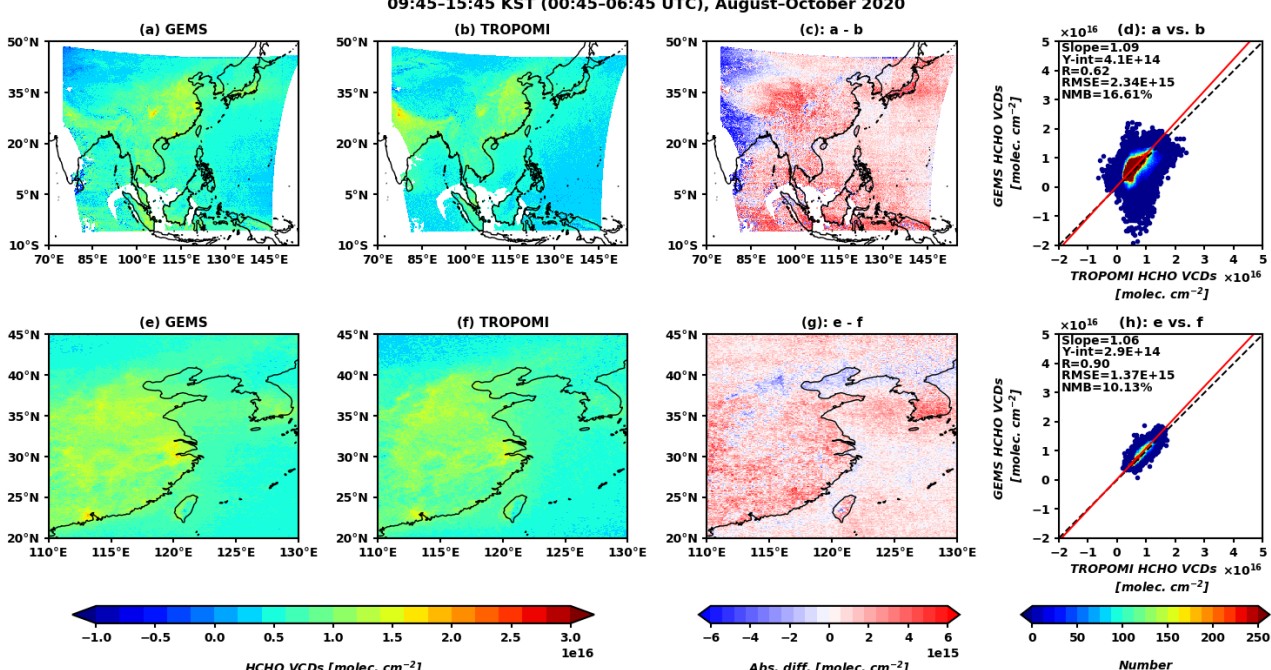


**Fig. 9. Mean HCHO VCDs from (a) GEMS and (b) TROPOMI products for TROPOMI overpass time (13:30, local time) during IOT (August–October 2020). Absolute differences between the GEMS and TROPOMI (a - b) are presented in (c), and their scatterplot is shown in (d) with the statistics. (e) to (h) in the second row are the same as (a) to (d) but restricted to East China and**
**the Korean peninsula (110–130° E, 20–45° N).**



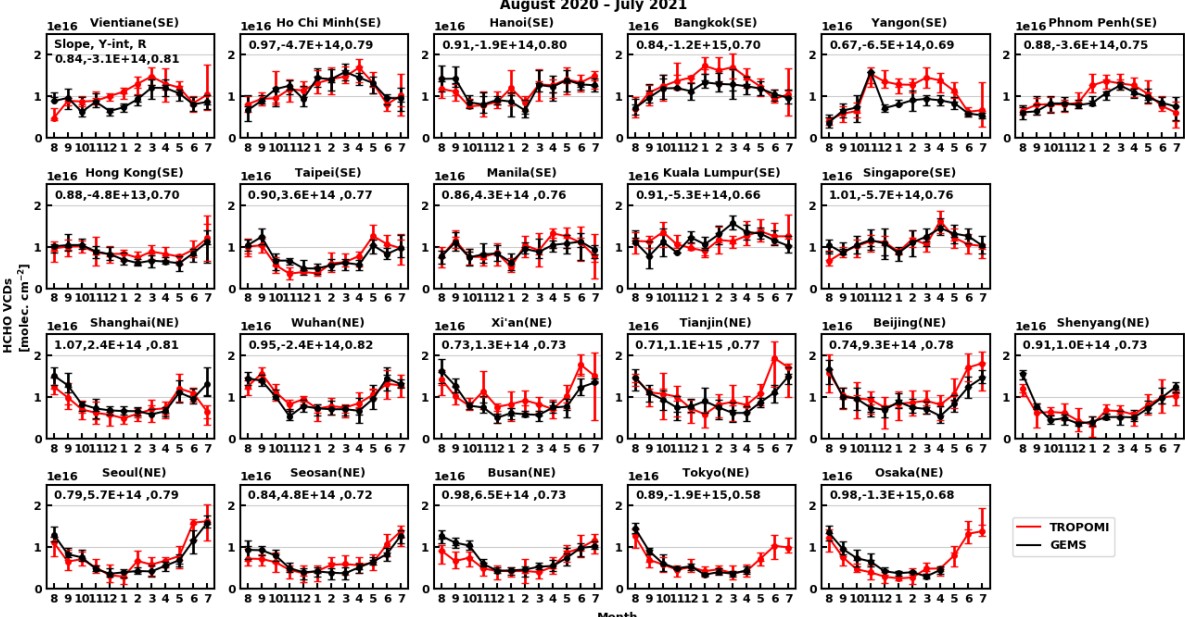

**Fig. 10. Comparison of the monthly mean HCHO vertical columns for GEMS and TROPOMI over 22 major cities in Southeast (SE) and Northeast (NE) Asia. Black and red solid lines represent GEMS and TROPOMI, respectively. Error bars are the first (25 %),**
**second (50 %), and third (75 %) quantiles of the columns and markers representing the means of each monthly dataset. Because GEMS is not observing eastern Japan during the TROPOMI overpass time after May 2021, VCDs over Tokyo and Osaka on June and July 2021 are missing.**



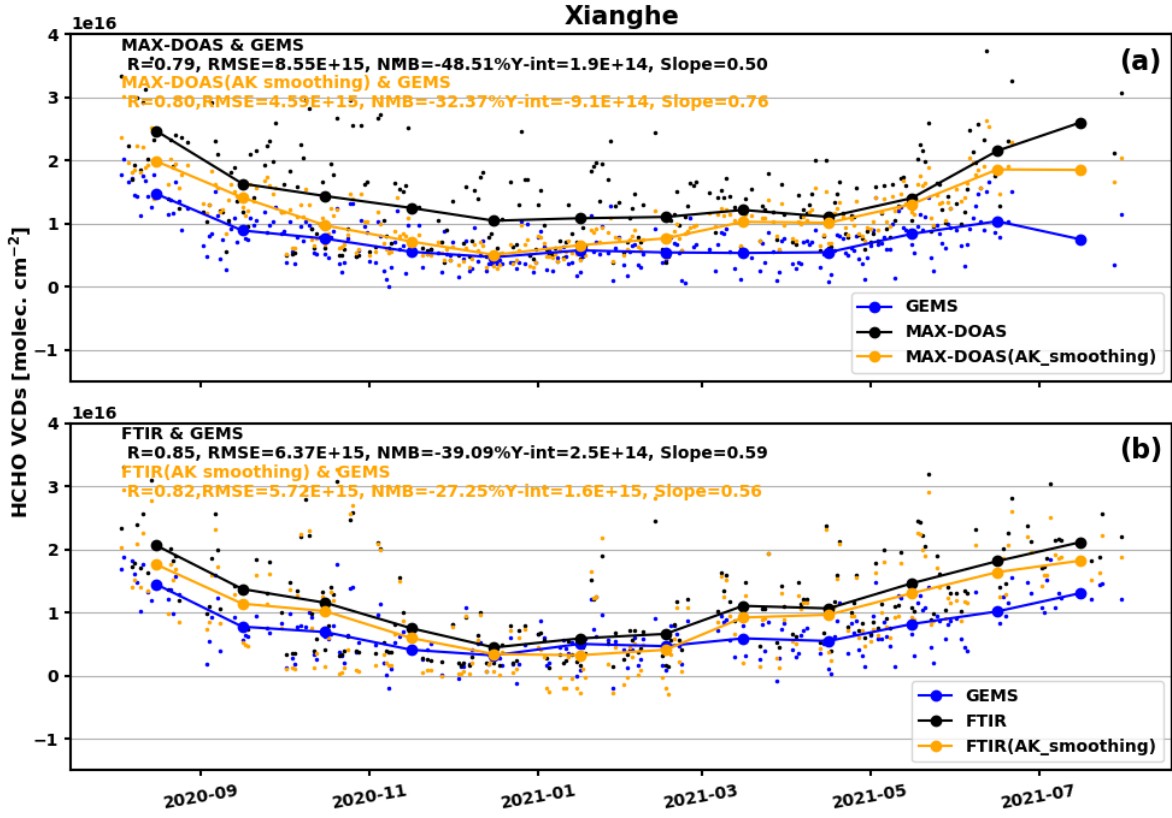

**Fig. 11.** Daily (small marker) and monthly (large marker) mean HCHO VCDs of GEMS (blue), MAX-DOAS (black), and MAX-DOAS with averaging kernel smoothing (orange) from August 2020 to July 2021 (a). (b): Same as (a) except for FTIR observation.





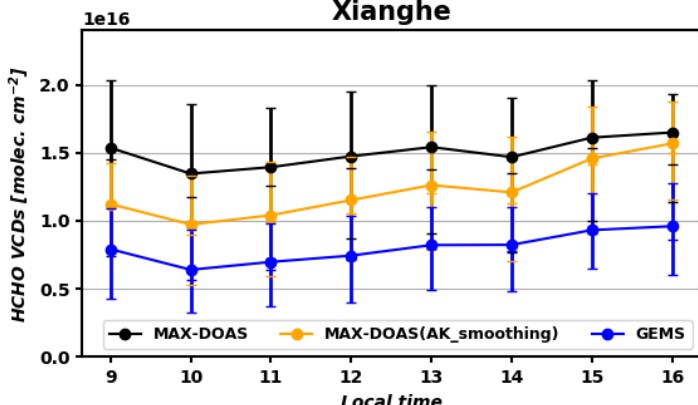

**Fig. 12. Hourly mean HCHO VCDs of GEMS (blue), MAX-DOAS (black), and MAX-DOAS with averaging kernel smoothing (orange) from August 2020 to July 2021. Error bars are the first (25 %), second (50 %), and third (75 %) quantiles of the columns.**
