# Peer review of "First evaluation of the GEMS formaldehyde product against TROPOMI and ground-based column measurements during the in-orbit test period"

_EGUsphere, 2023_

## Author Comment (AC1)

**Responses to Referee's Comments**

We appreciate the reviewers' constructive and insightful comments. We have carefully checked and addressed all comments and questions.

We have written the referee's comments in black and our responses to the comments in blue and italics. The revised sentences and paragraphs related to these comments are shown in red. Please note that the numbering of the figures and tables has been changed in the revised manuscript.

**Referee #1:**

The paper describes the comparison of the GEMS formaldehyde retrievals with those of TROPOMI and ground observations.
In addition sensitivity analyses are performed. It is a useful and complete manuscript but difficult to read sometimes, since it lacks precision in its formulations or because of the many grammar or style mistakes. After improving the writing style I would recommend it for publication.

1. Line 21: "...has monitored Asia." This is very vague. What has been monitored for Asia?

*We have added "atmospheric chemical compositions" to the sentence to clarify its meaning.*

L20: The Geostationary Environment Monitoring Spectrometer (GEMS) onboard GEO-KOMPSAT 2B was launched in February 2020 and has been monitoring atmospheric chemical compositions over Asia.

2. Line 29: "..over the latitude.." What is meant by this?

*We have deleted the sentence.*

3. Line 30: "..between the two is.." Between the two what ?

*We have modified the sentence by adding "GEMS and TROPOMI" instead of "the two".*

L26: We found that the agreement between the GEMS and TROPOMI was substantially higher in Northeast Asia (r=0.90), encompassing the Korean Peninsula and East China.

4. Line 96: "by applying air mass factor" => "by applying an air mass factor"

*We have made the necessary changes to this part.*

L93: Finally, post-processing performs background corrections using model columns from unpolluted clear areas and converts from SCD to VCD by applying an air mass factor (AMF) (Palmer et al., 2001).

5. Line 101: "Variables of GEMS HCHO Level 2 product" => "Variables of the GEMS HCHO Level 2 product"

*We have made the necessary changes to this part.*

L99: Variables of the GEMS HCHO Level 2 product are listed in Table S1.

6. Line 120-121: "..and Japan and could affect the background contributions to VCD" => "..and Japan, which can affect the background contributions to the VCD"

*We have made the necessary changes to this part.*

L122: The new reference sector partially includes polluted areas in East China, Korea, and Japan, which can affect the background contributions to the VCD.

7. Line 245 "who reported that cloud-free assumption" => "who reported that the cloud-free assumption"

*We have made the necessary changes to this part.*

L264: These results are highly similar to those of De Smedt et al. (2021), who reported that the cloud-free assumption could greatly reduce existing biases when comparing satellites over South Asian regions and perform less effectively in mid-latitude polluted areas.

8. Line 255 and 257 : what does %p mean?

*We intended to show the changes in two percentages as percent point (%p) unit, but this notation should be replaced by percent (%) in the current context.*

*We have made the necessary changes to this part.*

L274: Figure S7 shows the same comparison between GEMS and TROPOMI except for the $VCD_0$. In Southeast Asia, TROPOMI showed 10–15 % higher contributions of $VCD_0$ to VCDs than GEMS did, showing consistent correlation coefficients of r=0.51–0.73. However, in Northeast Asia, the difference in $VCD_0$ contributions between TROPOMI and GEMS widened by 70 %

9. Line 262-264: I think I understand the procedure you followed here for averaging the GEMS observations, but the description is very unclear.

*We have revised the sentence for improved readability. In addition, we have added a brief description of the temporal collocation methods.*

L282: We set a spatial grid of $0.4° \times 0.4°$ centered around the ground observatory and averaged GEMS observations within the grid. The effective size of the sampling grid was adopted from De Smedt et al. (2021), who determined a similar radius circle as the optimal value in the TROPOMI and MAX-DOAS HCHO comparisons. For temporal collocation, the MAX-DOAS and FTIR datasets were averaged to hourly data by a satellite observation time window of approximately 30 min.

10. Line 264-266: This is also an unclear description. Didn't you simply apply the averaging kernel of GEMS to the MAX-DOAS profile?

*We complemented an explanation regarding the necessity for vertical smoothing and their brief process.*

L294: The FTIR and MAX-DOAS products use different HCHO a priori profiles than the GEMS does, resulting in model dependencies when comparing their VCDs (Vigouroux et al., 2020; De Smedt et al., 2021; Kwon et al., 2023). To create inter-comparable datasets among GEMS and ground observations, we replaced the a priori profiles of the ground-based observations with those of GEMS interpolated by the same vertical grid based on

a smoothing method (Rodgers and Connor, 2003) and Eqs. 2 and 3 of Vigouroux et al. (2020).

11. Line 285: "GEMS pixel observing the MAX-DOAS station covers a much larger area, leading ..." => "The GEMS pixel covering the MAX-DOAS station has a large area, leading ..."

*We have made the necessary changes to this part.*

L317: The GEMS pixel covering the MAX-DOAS station had a large area, leading to diluted HCHO VCDs, especially when the observation area had a high HCHO concentration.

12. Line 290-291: ".. with the previous result from De Smedt et al. (2015), with the increasing trend.." => ".. with previous results from De Smedt et al. (2015), showing an increasing trend.."

*We have made the necessary changes to this part.*

L323: The diurnal variation of VCDs from the GEMS is consistent with previous results from De Smedt et al. (2015), showing an increasing trend of HCHO VCDs during the day.

13. Line 299: "was started by" => "was made by"

*We have made the necessary changes to this part.*

L340: The first geostationary satellite observation of HCHO was conducted by the GEMS, which enabled the investigation of the diurnal variability of HCHO over East Asia.

14. Line 323: " the large pixel size dilutes an error from the problematic area". Please rewrite this sentence.

*We have deleted the sentence regarding scene heterogeneity problem.*

15. In general, I think the conclusions section is a summary with a lot of repetition. I would expect also to see some statements about the precision and accuracy of the GEMS formaldehyde product in the conclusion.

*We have added a sentence regarding the precison and accuracy of GEMS HCHO.*

L358: We found high correlations between GEMS and TROPOMI HCHO VCDs and a good representation of seasonality with the regional characteristics of GEMS HCHO among the major cities, showing active emissions from biogenic and anthropogenic sources over East Asia.

---

## Author Comment (AC2)

**Responses to Referee's Comments**

*We appreciate the reviewers' constructive and insightful comments. We have carefully checked and addressed all comments and questions.*

*We have written the referee's comments in black and our responses to the comments in blue and italics. The revised sentences and paragraphs related to these comments are shown in red. Please note that the numbering of the figures and tables has been changed in the revised manuscript.*

**Referee #2:**

This paper describes upgrades to GEMS HCHO retrieval, and its comparison and evaluation versus TROPOMI, FTIR and MAX-DOAS observations. The upgrades to the "baseline" formaldehyde retrieval consist in fitting polarization sensitivity vectors and constructing radiance reference spectra using clear-sky pixels. The intercomparisons between GEMS and TROPOMI shows good correlation with a small bias of 10-16%. GEMS captures seasonal variations well, showing relatively high correlations with FTIR and MAX-DOAS ground-based observations. Overall, the paper is well written, even though it could benefit from an English language edit.

The paper has a clear structure and provides important information to understand and use GEMS formaldehyde retrievals. Addressing the questions that follow before publication would help to provide a more solid description of the retrieval upgrades and the comparison with correlative measurements. Its publication is therefore more than justified, providing important reference material for anyone working with GEMS formaldehyde retrievals.

1. The inclusion of polarization vectors and the modification of the radiance reference affects slant column density (SCD) retrievals. The discussion about the effects of these changes in the retrievals should focus on SCD. Describing them in terms of vertical column density (VCDs) compounds the changes on SCDs with AMFs (affected by the change in the a priori profiles) and they should be analyzed separately.

*We have updated the figures with differential slant column densities (dSCDs) below.*

**Questions regarding the calculation of the radiance reference and the background correction:**

2. Is the background correction computed as one North/South 1D vector? It would be very interesting to compare the background columns computed for different days using the old and new radiance reference / background corrections.

*We used North/South 1D vectors of monthly and zonal mean model background VCDs for the correction. We have added the comparison plot of annual mean background columns calculated from the old and new radiance reference sectors.*

[Figure]

**Fig. S2. Simulated annual mean HCHO background VCDs according to the reference sector (new: 120–150° E, old: 143–150° E).**

L123: The simulated background VCDs of the new reference sector exhibited 30 % higher values on average (Fig. S2).

3. Further details regarding the radiance reference calculation would be very useful, for example: What is the minimum number of pixels used to calculate the radiance reference? Why having larger signals (over clouds) is a problem? The intuition suggests that pixels affected by clouds could result in higher signals and therefore less noise. Are radiance references time of day dependent or is one radiance reference per day calculated?

*We have added detailed explanations regarding the construction of radiance references. In the case of the number of pixels for radiance reference, there are no specific criteria of minimum pixel numbers because the three-day mean radiance references generally provide sufficient pixels in the radiance reference sampling.*

L126: High reflectance conditions, such as thick clouds, can affect radiance references owing to the different magnitudes of radiances compared with typical background conditions.
L131: Because the observed light path varies significantly with the solar zenith angle (SZA), we used radiance references at the same observation time.

4. In line 130 it is suggested that radiance reference changes affect mostly high VCDs, yet I wonder if the radiance reference only affects the high value of the retrieved SCDs or is a uniform change? Figure 1, on top of the radiance reference itself could show the retrieved SCDs for one scan using the different radiance reference options. Also, it would be very interesting to include another figure illustrating the methodology used to calculate the background correction and its latitudinal dependency.

1. *Retrieved dSCDs using cloud-masked radiance references are mostly lower than those using radiance references without cloud masking, even though the amount of the reduction depends on latitudes.*

2. *We briefly described a formulation of background-corrected SCDs shown by Kwon et al. (2019)*

L137: Consequently, the use of the updated radiance references resulted in 10–40 % small differential SCDs (dSCDs) compared to those using radiance references without cloud masking over the entire scan domain.

L109: Background correction adds slant columns simulated by a chemical transport model to the reference sector to retrieve the slant columns as a function of latitude. Using Eq. (1), which is described in Kwon et

al. (2019), background correction was conducted using the model VCDs ($VCD_{CTM}$) and the satellite-derived AMFs ($AMF_0$) at a given latitude, resulting in a corrected SCD ($SCD_{corr}$) at each cross-track $i$ and along-track $j$.

$$SCD_{corr}(i, j) = dSCD(i, j) + AMF_0(lat) \, VCD_{CTM}(lat) \tag{1}$$

5. Another question that needs to be addressed here is the dependency of the retrieved SCDs with the radiance reference construction method. The paper does not provide enough details about how the radiance reference is constructed. For example, what is the effect of using the radiance reference from day i in day i+1? What about hour h on hour h+1?

*We tested the day-to-day variations of the radiance references. However, we did not include their results because it could cause duplications with Fig. 2. Using previous one- or two-day mean radiance references generally reproduces similar spectral and latitudinal distributions of the reference spectra. Therefore, the retrieved dSCDs are consistent, as shown in the figure below. Regarding the hour-to-hour dependency, we briefly described the necessity of using radiances of the same hour in L141.*

[Figure]

**Fig. R1. Same as for Fig. 1 except for cloud-masked radiance references of GEMS in 12:45 KST (03:45 UTC), 4–6 December 2020.**

**Questions regarding the inclusion of polarization vectors in the fitting:**

1. It will be very interesting to see the value of the polarization vector fitting parameter for a given scan? Does it have a bigger impact on some SZA/surfaces than others? What is the optical depth associated with the polarization vector pseudo absorption? As mentioned above, the fitting of the polarization vector affects the retrieved differential columns not the VCD. To avoid convolution of new fitting approach with AMF calculations and changes in the cloud algorithms the analysis of results affecting the spectral fit should look at

dSCDs only. This comment applies to figures 6, 7, 8...

*We have updated figures 6 and 8 with the dSCDs.*

*Indeed, polarization effects significantly contribute to retrieved HCHO dSCDs, especially under a high SZA and VZA. This phenomenon might be due to the behavior of polarization characteristics in the GEMS instrument. Following the statement of Choi et al. (2023), the degree of polarization error shows a maximum in the morning and afternoon, with a minimum error at noon. We have updated a figure including optical depths of the fitting parameters (Fig. 2 in the new manuscript). The optical depth for polarization sensitivity is shown in Fig. 2.*

[Figure]

**Fig. 5. Average time dependence of GEMS HCHO dSCDs with (a) and without (b) polarization correction and their relative differences (c) ((a–b)/b) during the IOT.**

[Figure]

**Fig. 7. Mean dSCDs from the GEMS HCHO algorithm with the optimized (a) and pre-launch fitting windows (b) for 13:45 KST (04:45 UTC) on 2 September 2020, with their absolute differences (a–b) and scatter plot.**

[Figure]

**Fig. 8. Mean HCHO dSCDs from the GEMS using measured radiance references (a) and irradiance with de-striping (b) for 09:45–15:45 KST (00:45–06:45 UTC) during the IOT (August–October 2020), with their absolute differences (a–b) and scatter plot.**

[Figure]

**Fig. 2. Fitted optical depths (black solid line) and optical depths plus the fitting residuals (red solid line) of the operational GEMS HCHO retrieval algorithm in Northern Myanmar at 12:45 KST (03:45 UTC), 3 August 2020.**

2. What about the correlation between the retrieved polarization parameter and HCHO fitting?

*In the high SZA and VZA conditions, polarization is more sensitive to the spectral fitting. However, the effects on the spectral fitting are minimized in the lower zenith angles, indicating less correlation between the retrieved polarization parameter and HCHO.*

**Questions associated with the discussion of the sensitivity tests:**

1. Line 179 and figure 8: The paper should compare dSCDs with SCDs, since the VCD contains information about the AMF, imposing a priori information that most likely increases the correlation between both retrievals above the one expected if only dSCD and SCDs where considered.

*We acknowledge the necessity of comparing dSCDs to separate the impact of AMF. As mentioned above, we have modified this part to compare dSCDs instead of VCDs in Fig. 7.*

[Figure]

**Fig. 7. Mean dSCDs from the GEMS HCHO algorithm with the optimized (a) and pre-launch fitting windows (b) for 13:45 KST (04:45 UTC) on 2 September 2020, with their absolute differences (a–b) and scatter plot.**

L184: Figure 7 compares the retrieved HCHO dSCDs under the pre-launch and optimized fitting windows at 13:45 KST (04:45 UTC) on 2 September 2020. HCHO dSCDs using an optimized fitting window (Fig. 7a) presented 10–30 % higher values than those of the pre-launch fitting window (Fig. 7b) but showed consistent spatial correlations (r=0.95) and good representations of local emissions over East China and the Korean Peninsula.

2. What is the behavior below 40N? Looking at figure 8 looks like if the bias is gradual?

*We found that the previous version of the background profile was applied in the de-striped irradiance reference VCDs in the comparison. The new background profile was used in the de-striping process, leading to consistent spatial distributions (Fig. R2).*

[Figure]

**Fig. R2. Same as for Fig. 7 except for the VCDs comparison between the default (a) and irradiance reference retrieval (b).**

*Per your comments, we showed the comparison of dSCD between the irradiance and radiance reference retrieval (Fig. 8 in the new manuscript). The results showed consistent spatial distributions with correlation coefficients of 0.97. However, since the fitting RMS and uncertainties showed a substantially higher value of irradiance reference retrieval, the use of irradiance references still needs an elaborate de-striping algorithm to address the high fitting residuals and uncertainties.*

L193: HCHO dSCDs using irradiance spectra were in good agreement (r=0.97) with those using radiance references but were 20–50 % higher on the west side of the scan domain (< 100° E). In addition, HCHO products using the irradiance reference showed 10–50 % higher fitting RMS (approxiamtely $2.5 \times 10^{-3}$) and random uncertainties (approxiamtely $8 \times 10^{15}$ molecules cm$^{-2}$) than those using the radiance reference. Based on the lower fitting RMS and uncertainties, a radiance reference was used as the reference spectrum in the operational retrieval.

[Figure]

**Fig. 8. Mean HCHO dSCDs from the GEMS using measured radiance references (a) and irradiance with de-striping (b) for 09:45–15:45 KST (00:45–06:45 UTC) during the IOT (August–October 2020), with their absolute differences (a–b) and scatter plot.**

3. The higher RMS using irradiances points towards unaccounted spectral signals in the model of the fit (as expected) that are accounted for when using the radiance reference.

*We have added a specific reason for the high fitting RMS in the irradiance reference retrieval.*

L197: The elevated fitting RMS and uncertainties in the irradiance reference retrieval may be due to the unaccounted spectral signals in the spectral fitting process, which were addressed when employing the radiance reference.

4. Line 183: Results using radiance and irradiance are almost not identical. The differences are significant and, depending on the location, important. Please review this statement.

*We acknowledge our misconception that was analyzed based on the high correlation coefficient. We have deleted the statement of "identical results" and added a cause for using radiance references in terms of fitting RMS and uncertainties.*

L193: Based on the lower fitting RMS and uncertainties, a radiance reference was used as the reference spectrum in the operational retrieval.

**Questions regarding the comparison with TROPOMI observations:**

1. It will be very interesting to break down the comparison between TROPOMI and GEMS in terms of SCDs, AMFs and VCDs to try to understand the performance of those two retrieval steps separately. Even more interesting would be to compare VCDs pre and post reference sector background correction.

*Thank you for the comments. It would be informative to compare the AMFs and SCDs. However, since these variables highly depend on observing the geometries of the satellite instruments, it is difficult to directly compare these variables due to the different geometric angles between GEMS and TROPOMI. Instead, we separated the impact of cloud properties in AMF calculations, as shown in Fig. S3, which minimizes the discrepancies in cloud fitting windows between GEMS and TROPOMI.*

2. Line 250: this is an interesting discussion if done with care. First it is important to establish how much of the latitudinal variability comes from the background correction for GEMS (for TROPOMI there is no dependency on the background correction but there is also a bias correction, is that considered?). Second, it is necessary to understand which percentage of the reported VCD is coming from the background correction? Finally, how large is GEMS background correction variability as function of SZA and day of year/season.

*As described in Kwon et al. (2019), GEMS has a spatial dimension from North to South, and radiance references are made at each spatial pixel. Therefore, latitudinal biases could be minimized by using radiance references because measurements taken from each latitude already include certain factors that cause systematic biases in spectral fitting. Regarding the contribution of the background corrections to VCDs, we presented the ratio of VCD and $VCD_0$ (VCDs without background corrections) to investigate the background contributions of GEMS and TROPOMI from August 2020 to July 2021 in Fig. S7, showing the seasonal variations of background correction variability.*

3. Line 192: Please add detail about the filtering criteria associated with QA<0.5 (particularly for what it refers to cloud fraction and SZA) since those two parameters are used to filter out GEMS. What does it mean FinalAlgorithmFlag=0 in GEMS?

*We have added brief descriptions of the quality flags used in this study.*

L207: We filtered out unqualified values of TROPOMI HCHO VCDs using the "Quality Assurance (QA)"

variable (QA < 0.5), which is a recommended limit determined from observation conditions and other retrieval flags. For GEMS, we used the operational Level 2 HCHO product (version 2.0) and selected pixels in a "good" quality flag (FinalAlgorithmFlags = 0), which filters out pixels with high fitting residuals by using median absolute deviations (MADs) derived from the fitting RMS in a scan domain (fitting RMS < median (fitting RMS) + 3 × MAD (fitting RMS)). In addition, pixels with cloud radiance fractions less than 0.4 and low geometric angles (SZA < 70° and VZA < 70°) were used for the validation.

4. Line 194: Does this mean that only GEMS pixels with LT of 13:30 are used? I guess the answer is not since TROPOMI overpass time off nadir is different from 13:30 LT. Please provide further details.

*We have added the detailed collocation method for the TROPOMI and GEMS comparison.*

L207: GEMS pixels were temporally collocated using the TROPOMI observation time within a ± 15 min time window.

**Questions regarding the comparison with FTIR and MAX-DOAS**

1. Are there plans to make comparisons in other locations?

*We plan to compare GEMS HCHO VCDs with MAX-DOAS observations in Korea and Japan. In addition, a ground-based Pandora Asia Network (PAN) is currently being constructed over Asia, and data from Pandora will be used to validate GEMS products. We also plan to use the column measurements from the recent Satellite Integrated Joint Monitoring of Air Quality (SIJAQ) and the upcoming Airbourne and Satellite Investigation of Asian Air Quality (ASIA-AQ) campaigns, which conducts a holistic examination of regional air qualities in East Asian countries.*

2. Line 282: Please, place the results of this GEMS comparison with FTIR and MAX-DOAS in the context of Vigouroux et al., 2020 and de Smedt et al., 2021 with TROPOMI information. Are correlations and biases similar? Please quantify. This would be useful since given the preceding discussion regarding GEMS and TROPOMI comparisons.

*We have compared the daily and monthly mean HCHO VCDs of GEMS and TROPOMI against that of MAX-DOAS during the TROPOMI overpass time but have not placed this figure in the manuscript because our focus was to examine the diurnal variability of GEMS HCHO. In addition, the monthly behavior of TROPOMI HCHO VCDs was already compared with GEMS in Fig. 10. However, we acknowledge the necessity of general evaluation results in the satellite against ground-based observations. We have briefly described the comparison results in the manuscript and have added a figure below in the supplementary information.*

L289: First, we compared the daily and monthly mean HCHO VCDs of GEMS and TROPOMI with those of MAX-DOAS and FTIR during the TROPOMI overpass time (1:30 pm, local time), as shown in Fig. S8. GEMS (r=0.74) and TROPOMI (r=0.73) presented good correlations but showed negative NMBs (GEMS=−45.22 %, TROPOMI=−34.7 %) with MAX-DOAS. Similar statistics are presented in the case of the comparison with FTIR, except for the lower value of NMBs, showing correlation coefficients of r=0.85 and 0.63 and NMBs of −37.7 % and −31.37 % for GEMS and TROPOMI, respectively. However, the FTIR products from October 2020 to January 2021 had insufficient data points that overlapped with the TROPOMI overpass time.

[Figure]

**Fig. S8.** Time series of daily (small marker) and monthly (large marker with solid line) mean vertical columns for GEMS (blue) and TROPOMI (red) against (a) MAX-DOAS (black) and (b) FTIR (black) for the TROPOMI overpass time (13:30, local time). GEMS and TROPOMI HCHO VCDs were directly compared with MAX-DOAS and FTIR without averaging kernel smoothing and a priori substitution.

3. Line 285: Does Xianghe have a forested area that could explain large biogenic emissions of isoprene; are they of anthropogenic origin? It would be great to provide some context.

*Xianghe is predominantly agricultural land with a few residential areas with high isoprene emissions. We have referred to a study investigating the driving emission sources of isoprene in that area.*

L312: Xianghe is a suburban area that primarily consists of agricultural areas with partial residential areas where large isoprene emissions occur (Xue et al., 2021).

4. Line 290: Is the diurnal evolution of GEMS columns linked to increased formaldehyde or does it has to do with the development of the boundary layer during the day? In the early hours of the day, even if there is a large HCHO column it would be near the surface. GEMS sensitivity to that early morning shallow boundary layer is very limited. Most variability of the columns (spatial) at those hours I would assume is associated with a priori information of the AMF calculation unless proven otherwise by showing heterogeneity in the slant columns. What is the correlation at different hours of the day?

*We acknowledge the limited sensitivity of GEMS observations and the possibility of the contribution of a priori profiles in the early morning. We compared diurnal variations among HCHO dSCDs, VCDs using Geometric AMF (GAMF), and model-derived VCDs from a priori profiles to separate the impact of a priori profiles to GEMS VCDs.*

L331: When a shallow boundary layer in the early morning restricts HCHO concentrations to the surface, GEMS can cause large uncertainties in the observation of HCHO columns owing to its limited sensitivity. In this

scenario, the a priori profiles can dominantly contribute to the calculation of VCDs. To examine the impact of a priori profiles in the morning, we recalculated the VCDs using the dSCDs divided by the geometric AMF (GAMF) (Palmer et al., 2001): Figure S12 shows the diurnal variations in the HCHO dSCDs, VCDs using the GAMF, and model VCDs from the GEMS a priori profile, averaged from August 2020 to July 2021. Both dSCDs and VCDs using GAMF showed consistent diurnal variations with the a priori profiles, implying that the GEMS observes the morning time variabilities well without using the a priori profiles. Further studies on the possible uncertainties of the a priori profile simulations from the model should be conducted (Yang et al., 2023).

[Figure]

**Fig. S12. Diurnal variations of median HCHO column concentrations from August 2020 to July 2021 in Xianghe: GEMS VCDs using geometric AMF (GAMF) without background corrections (black), GEMS dSCDs (blue), and model VCDs from GEMS a priori profiles (red).**

**Minor/specific comments:**

1. Abstract. The description of the comparison between GEMS and TROPOMI is interrupted by two sentences introducing the use of polarization vectors and reference spectrum calculations. Maybe it would be more logical to move these two sentences at the end of the discussion concerning GEMS and TROPOMI comparisons.

*We have revised a sequence of sentences regarding the updates and validation parts of the algorithm.*

L20: The Geostationary Environment Monitoring Spectrometer (GEMS) onboard GEO-KOMPSAT 2B was launched in February 2020 and has been monitoring atmospheric chemical compositions over Asia. We present the first evaluation of the operational GEMS formaldehyde (HCHO) vertical column densities (VCDs) during and after the in-orbit test period (IOT) (August–October 2020) by comparing them with the products from the Tropospheric Monitoring Instrument (TROPOMI) and Fourier-Transform Infrared (FTIR) and Multi-Axis Differential Optical Absorption Spectroscopy (MAX-DOAS) instruments. During the IOT, the GEMS HCHO VCDs reproduced the observed spatial pattern of TROPOMI VCDs over the entire domain (r=0.62) with high biases (10–16 %). We found that the agreement between the GEMS and TROPOMI was substantially higher in Northeast Asia (r=0.90), encompassing the Korean Peninsula and East China. GEMS HCHO VCDs captured

the seasonal variation in HCHO, primarily driven by biogenic emissions and photochemical activities, but showed larger variations than those of the TROPOMI over coastal regions (Kuala Lumpur, Singapore, Shanghai, and Busan). In addition, GEMS HCHO VCDs showed consistent hourly variations with MAX-DOAS (r=0.78) and FTIR (r=0.86) but were 30–40 % lower than ground-based observations. Different vertical sensitivities of the GEMS and ground-based instruments caused these biases. Utilizing the averaging kernel smoothing method reduces the low biases by approximately 10 to 15 % (normalized mean bias (NMB): −47.4 % to −31.6 % and −38.6 % to −26.6 % for MAX-DOAS and FTIR, respectively). The remaining discrepancies are due to multiple factors, including spatial collocation and different instrumental sensitivities, requiring further investigation using inter-comparable datasets.

2. Line 57: "adopted" implies the instrument could have chosen the resolution. Maybe "have much finer" is a better description if combined with "allowing to observe" ("have adopted much finer… allowing to observe local pollution plumes…"

*We have made the necessary changes to this part.*

L52: Subsequent LEO satellites, including the Ozone Monitoring Instrument (OMI), Tropospheric Monitoring Instrument (TROPOMI), Global Ozone Monitoring Experiment 2A (GOME-2A), and Ozone Mapping and Profiler Suite (OMPS) nadir mapper, have substantially finer spatial resolutions (approximately 5.5 × 3.5 km–80 km × 40 km) enabling the observation of local pollution plumes and the provision of observational constraints for biogenic and anthropogenic sources globally (Veefkind et al., 2012; De Smedt et al., 2015, 2021; Li et al., 2015; González Abad et al., 2016; Levelt et al., 2018; Nowlan et al., 2023; Kwon et al., 2023).

3. Line 63: "the presence of clouds"

*We have made the necessary changes to this part.*

L59: However, limited by the overpass time, these LEO satellites provide observations at most once daily, which can be significantly compromised by the presence of clouds, especially in East Asia.

4. Line 76: What is the definition of "co-added" pixels (how many)?

*We have added specific pixel numbers (2 × 2 or 4 × 4) integrated with the co-addings. Detailed co-adding information for the GEMS products is presented in Sect. 2.*

L71: The Geostationary Environment Monitoring Spectrometer (GEMS), launched on 19 February 2020 by the Korean Ministry of Environment, has begun hourly observations of trace gases (NO2, SO2, O3, HCHO, and CHOCHO) and aerosols with 3.5 km × 8 km pixels or co-added pixels (2 × 2 or 4 × 4) over Seoul, Korea (Kim et al., 2020).

5. Line 91: "consisting of a three steps", remove processes.

*We have made the necessary changes to this part.*

L89: Kwon et al. (2019) described the GEMS HCHO retrieval algorithm (v0.3), which consists of three steps: pre-processing, spectral fitting, and post-processing.

6. Line 95: suggested change "by" to "using" "…corrections using model columns from unpolluted clear areas"

*We have made the necessary changes to this part.*

L93: Finally, post-processing performs background corrections using model columns from unpolluted clear areas and converts from SCD to VCD by applying an air mass factor (AMF) (Palmer et al., 2001).

7. Line 111: suggested change "is to" to "…the background correction adds slant…"

*We have made the necessary changes to this part.*

L109: Background correction adds slant columns simulated by a chemical transport model to the reference sector to retrieve the slant columns as a function of latitude.

8. Line 117: suggested change " …GEMS cannot sufficiently obtain clean…" to "…GEMS cannot obtain sufficient clean…"

*We have made the necessary changes to this part.*

L120: Therefore, GEMS cannot obtain sufficient clean pixels from the Pacific Ocean on an hourly basis.

9. Line 136: Are these residuals and fitting uncertainties characteristic of retrievals at any time of the day/location or there is a diurnal dependency of the uncertainties with SZA and the brightness of the scene?

*The fitting residuals and uncertainties are generally high under high SZA and HCHO concentrations. Figures R3 and R4 show the diurnal variation in retrieved fitting RMS and uncertainties of GEMS HCHO, respectively. They show increasing features of their values from 10:45 KST to 15:45 KST and are consistent with the HCHO diurnal variation, which is also continuously increasing. We have added a sentence describing diurnal variations of fitting residuals and uncertainties.*

[Figure]

**Fig. R3. Diurnal variations of GEMS HCHO fitting uncertainties for 3 August 2020.**

[Figure]

**Fig. R4. Same as for Fig. R3 except for the fitting RMS.**

L147: GEMS also showed high residuals (approximately $1.5 \times 10^{-3}$) and uncertainties (approximately $1.0 \times 10^{16}$ molecules $cm^{-2}$) under high SZAs, rendering the spectral fitting more uncertain.

10. Line 142: This sentence is confusing. Is the LUT of scattering weights recalculated using GEOS-Chem at higher resolution? Which profiles (ozone, temperature...). What I think is going on here is that the HCHO a priori profiles have been updated but the LUT remains the same. It could be interesting to briefly remind the reader the parameters of the LUT table.

*The LUT of scattering weights from Kwon et al. (2019) remains, but the a priori profiles have been updated. Indeed, the scattering weight LUT, with substantially finer resolutions and recent emission inventories, can improve the AMF calculations. However, in this study, we focused on the impact of changing a priori profiles as the local emission rates over East Asia are updated. We have added a few details for the configurations of the scattering weight and a priori profiles.*

L150: The AMF look-up-table of Kwon et al. (2019) consists of pre-calculated scattering weights based on monthly mean trace gas ($O_3$, $NO_2$, $SO_2$, and HCHO) and temperature profiles simulated from GEOS-Chem, with a spatial resolution of $2° \times 2.5°$, and vertical shape factors calculated from identical a priori profiles. In this study, we only updated the vertical shape factors from the new monthly mean hourly a priori profiles simulated by GEOS-Chem with a substantially finer spatial resolution of $0.25° \times 0.3125°$ and up-to-date anthropogenic and biomass burning emission inventories in Asia (Table 2).

11. Line 170: Would it be possible to quantify the change in the retrieved SCDs between the pre-launch and "optimized" fitting window? Adding a plot showing the SCDs for one scan using both fitting windows and their difference would be interesting.

*As shown above, we have added a comparison plot for dSCDs between the pre-launch and optimized fitting windows in Fig. 7.*

12. Line 178: What is the meaning of "with background correction"? If I'm not mistaken the use of the irradiance instead of a radiance makes the background correction unnecessary.

*When we used the de-striping method to the retrieved slant columns, the median value of the unpolluted background pixels was subtracted from all pixels in the cross-track. Therefore, we had to add the subtracted background SCDs from the model. To avoid misconceptions, we deleted "with background correction" in the sentence.*

L191: We performed a de-striping process (Lerot et al., 2021) by subtracting the median values of each cross-track.

13. Line 208: What does it mean that the value of cloud fraction increases exponentially for SZA? Is the cloud fraction proportional to e^SZA?

*We have modified the sentence by changing it to "value" instead of "deviations," following Kim et al. (2021), who present that the deviations for cloud fractions exponentially increase at a higher SZA.*

L228: The uncertainty of the cloud fraction retrieval increases exponentially for SZA values above 40° and becomes significant above 60° (Kim et al., 2021).

14. Line 215: Figure 9, could the authors shows the effect of using two different radiance references (one calculated with FW the other one with the nominal scan)?

*HCHO VCDs retrieved using FW radiance references show 5–8 % higher values than those of nominal scans. The primary cause of the higher VCDs in FW radiance references is the background corrections, including higher HCHO concentrations over the narrower reference sector. We are investigating the impact of background concentrations on the retrieved vertical columns.*

[Figure]

**Fig. R5. Same as for Fig. 9 except for the radiance references sampled in the FW scan area during the nominal scan schedule.**

[Figure]

**Fig. R6. Same as for Fig. R5 except for the radiance references sampled in the nominal scan area.**

15. Line 222: This is not the only source of difference in the AMF calculations? A detail analysis should breakdown the contributions from surface properties, clouds, a priori profiles and scattering weight LUT. Lorente et al., 2016 paper could be very illustrative of the different contributions and guide further analysis.

*Indeed, various factors contribute to AMF calculations, which can cause the differences between GEMS and TROPOMI. However, this paragraph aims to introduce the cloud contributions from using cloud-free AMF in the GEMS and TROPOMI comparison. By comparing the key features of VCDs$_{cf}$ with the results of De Smedt et al. (2021), we evaluated the necessity of the cloud-free assumption to build better comparison conditions between the two satellite products. We mentioned Lorente et al. (2017) to describe the necessity of reminding other key factors in AMF calculations in the sentences.*

L240: Several factors, including cloud properties, surface albedo, and trace gas profiles, contribute to AMF calculations. We focused on the differences in the cloud properties between GEMS and TROPOMI. GEMS and TROPOMI use the observed radiances at different wavelength bands to derive cloud properties (O4 at 477 nm for GEMS vs. the O2–A band at 760 nm for TROPOMI), retrieving the different physical interpretations of cloud fractions and cloud pressures (Kim et al., 2023; Loyola et al., 2018).

L247: To exclude cloud dependency on the HCHO AMF in the comparison between GEMS and TROPOMI, we defined cloud-free VCDs (VCDs$_{cf}$) by applying AMFs under a cloud-free assumption, which was introduced by Lorente et al. (2017) and De Smedt et al. (2021).

16. Line 240: Adding information about collocation statistics for each city and month would be helpful to understand the significance of the comparison.

*We have added a supplementary figure for each city's sampled pixel numbers for the GEMS and TROPOMI.*
L263: The total number of pixels sampled (Fig. S5) over Japan (Tokyo: 60, Osaka: 76) was nearly one-third of the overall mean pixel count for all cities (mean pixel number: 200.4).

[Figure]

**Fig. S5. Total sampled pixels of GEMS and TROPOMI HCHO VCDs by the regions presented in Fig. 10 during the TROPOMI overpass time from August 2020 to July 2021.**

17. Line 320: Problems associated with scene heterogeneity are first mentioned in the conclusions of the paper and not discussed in its body. Either no mention it at all or please add a section explaining how it affects the retrievals, when and where?

*We have deleted the sentences regarding the possibility of scene heterogeneity.*

18. Table 1:

I understand that the paper does not discuss the construction and use of a common mode (mean residual) so maybe this question is outside the scope of this paper, but I wonder what is the dependency of the common mode with time of the day.

*We have compared the diurnal variations of dSCDs and fitting RMS with and without the common mode; however, there were no significant changes (< 1 %) for the entire day. Therefore, the deviations in dSCDs and fitting residuals caused by common mode contributions are negligible in the current scenario.*

Are the scattering weights precalculated for each 2x2.5 grid? What is the point of doing so, accounting for different ozone and aerosol profiles? The surface albedo is one of the parameters so don't see the benefit? What is the number of the LUT vertical layers?

*We used a scattering weight LUT consisting of 47 vertical layers, which directly followed the input model data from GEOS-Chem $2° \times 2.5°$ simulations. According to the AMF sensitivity tests to scattering weight parameters (cloud top pressure, cloud fraction, albedo, etc.) presented by Kwon et al. (2019), we considered the scattering weight LUT to be well-established and focused on updating a priori profiles in AMF calculations. We acknowledge that the surface reflectance can affect the scattering weight calculations. Recently, the GEMS level 2 background surface reflectance product, which considers BRDF modeling for surface reflectance retrieval, became available. Further investigations concerning the scattering weight calculations with the use of the GEMS surface reflectance product will be conducted in future studies.*

19. Figure 1: Which fitting window is used by TROPOMI retrievals (the only one relevant for this study)? Why show the others? Second, cutting the formaldehyde feature at 330 nm by starting at 329.3nm instead of 328.5 nm is counter intuitive. Showing correlation with other fitting parameters is crucial to understand the benefit of starting at 329.3 nm. What is the point of showing a normalized solar spectrum here? Likewise with the ring spectrum. This figure would be more useful if broken into panels showing the typical optical depth of each parameter within the fitting window. This figure is quite misleading giving the impression of very strong formaldehyde signals when it is the weakest signal fitted in this fitting window together with O4.

*We intended to introduce the general conventions of the HCHO fitting windows used in the representative LEO satellite sensor. To avoid misconceptions, we have deleted Fig. 1 and modified Fig. 3 to display the optical depths of all chemical species fitted in the retrieval algorithm.*

20. Figure 2: As mentioned before this figure would give a better understanding of the impact of the radiance reference if it included an example of the SCDs retrieved for one scan using different radiance references (old vs clear sky).

*We have added the differences in dSCDs retrieved by the three-day mean cloud-masked radiance references compared to those of the default in Fig. R7. However, we speculate that the inclusion of additional panels provides excess information and can lead to potential misconceptions for readers. To enhance the clarity of the*

*message in the figure, we decided not to present the difference in dSCDs in Fig. 1 (in the new manuscript).*

[Figure]

**Fig. R7. Latitudinally averaged radiance references of GEMS (03:45 UTC (12:45 KST), 6 December 2020): With cloud masking (cloud radiance fraction > 0.4) (a), without cloud masking (b), and cloud masking with three-day mean radiances (c). Absolute differences of dSCDs using (a) and (b) radiance references (d). The shadings are radiance spectra. The radiance spectra in the 1233–1241 cross tracks have bad L1C quality flags.**

21. Figure 3 could be more interesting if it showed the optical depth of other parameters considered in the fitting (such as o3, Ring, BrO, and o2-o2). Not a very important change. As it stands now, if doesn't bring much information to the analysis.

*We have added other species' optical depths and their fitting residuals in Fig. 2 (See Q1)*

L139: Figure 2 shows the fitted optical depths as a function of the wavelength for a specific pixel in Northern Myanmar at 12:45 Korean Standard Time (KST) on 3 August 2020. The black solid line represents the fitted optical depth of the chemical species, and the red solid line represents the optical depth along with the fitting residuals.

22. Figure 5 suggestion; plotting some fitting residuals (including the polarization vector vs. not) would be very convincing if the dramatic signal around 350 nm shows clearly.

*We show the optical depth of polarization sensitivity in Fig. 2. Polarization does not show the steep variations of fitting residuals around 350 nm.*

23. Figure 6: Is the white area in the southern part of the plot moving towards the west as the day progresses the glint cone? As expected, the effect of the polarization depends greatly on the geometry of the retrieval. It seems that backward scattering results in higher polarization signal. Is that correct? Is the fitting window used in the top and middle row retrievals the same?

*1. Yes, the white circle area over the southern part of the plot shows sun glint pixels masked by the quality flag*

*due to the high retrieval uncertainties.*

*2. Yes, backward scattering contributes to the high polarization signal. Following the statement of Choi et al. (2023), polarization error is a function of satellite–sun geometry. However, the interpretation of GEMS polarization characteristics needs to be constrained because Choi et al. (2023) did not fully cover their relationship with other GEMS scan areas after the IOT (Full West, Full Central, Half East, and Half Korea). Therefore, further investigations should be conducted to evaluate the polarization characteristics in the other scan area.*

*3. Yes, we used the identical fitting window for panels (a) and (b).*

24. Table S1:

While the cross-track and along-track jargon is meaningful to those familiar with LEO instruments similar, it does not really apply to GEO satellites since they are not moving on a orbit with a pre-determine track. Don't know if there is a better terminology to better describe GEO observations.

*We acknowledge that the cross-track and along-track do not efficiently represent the geolocations in geostationary orbit. However, we decided to use this jargon to create a consistent physical interpreation for understanding the conventional retrieval process from the same point of view as LEO satellites.*

Is there some other documentation explaining the contents of these variables? Particularly important to use the product would be to specify the meaning of the different flags. If so, it could be very useful to provide a link to the document or the page where this "file document format" document resides.

*The initial version of the algorithm theoretical basis document is available at "https://nesc.nier.go.kr/ko/html/satellite/doc/doc.do". However, since these documents are outdated (v0.3), we did not refer to this literature to avoid misleading the readers.*

What are the different degrees of convergence in the fit?

*The degree of convergence indicates a fitting score of the HCHO differential slant columns. The convergence flag is primarily defined by the difference between observed and modeled radiance spectra during the spectral fitting. We have added a description of the convergence flags in Table S1.*

**Table S1. Summary of the data field variables archived in the GEMS Level 2 HCHO product. Layer, spatial, and image represent the number of vertical layers of the a priori profile, cross-track, and along-track of the GEMS observation, respectively.**

| Variable name | Description | Unit | Dimensions |
|---|---|---|---|
| AirMassFactor | AMF | unitless | spatial × image |
| ClearAirMassFactor | AMF (cloud-free condition) | unitless | spatial × image |
| AMFCloudFraction | Cloud radiance fraction | unitless | spatial × image |
| AMFCloudPressure | Cloud top pressure | hPa | spatial × image |
| AMFDiagnostic | Diagnostic flags in AMF calculation | unitless | spatial × image |

| AMFSurfaceLER | Surface Lambertian-equivalent reflectivity | unitless | spatial × image |
|---|---|---|---|
| ColumnAmount | HCHO VCD | molecules cm$^{-2}$ | spatial × image |
| ColumnUncertainty | Random uncertainty of the HCHO VCD | molecules cm$^{-2}$ | spatial × image |
| dSCD | Retrieved SCD before background correction | molecules cm$^{-2}$ | spatial × image |
| FitConvergenceFlag | Degree of convergence of the spectral fitting determined by the fitting score of the HCHO DSCD | unitless | spatial × image |
| FittingRMS | Fitting root mean square error | unitless | spatial × image |
| GasProfile | HCHO a priori profile | molecules cm$^{-3}$ | layer × spatial × image |
| Layer | Pressure profile | hPa | layer × spatial × image |
| ScatteringWeight | Scattering weight | unitless | spatial × image |
| ClearScatteringWeight | Scattering weight (cloud-free condition) | unitless | spatial × image |
| FinalAlgorithmFlags | Final algorithm flags | unitless | spatial × image |

26. Figure S1: Where is the eastern boundary of the full central scan (pink)? Maybe it could be useful to use a discontinuous line on the eastern edge to show where it is, since I assume it overlaps with full western or half Korea eastern edges?

*We have updated the line style of the full central scan domain to be a dashed line.*

[Figure]

**Fig. S1. Operational scan domain of GEMS (adapted from Kwon et al., 2019): half eastern scan (blue), half Korea scan (black), full central scan (dashed magenta), full western scan (cyan), and the GEMS location (green star). Shaded areas (120–150° E) represent regions for radiance references and the common mode.**

**References**

Choi, H., Liu, X., Jeong, U., Chong, H., Kim, J., Ahn, M. H., Ko, D. H., Lee, D., Moon, K.-J., and Lee, K.-M.: Geostationary Environment Monitoring Spectrometer (GEMS) polarization characteristics and correction algorithm, Atmospheric Measurement Techniques Discussions, 2023, 1–33, 2023.

De Smedt, I., Pinardi, G., Vigouroux, C., Compernolle, S., Bais, A., Benavent, N., Boersma, F., Chan, K.-L., Donner, S., and Eichmann, K.-U.: Comparative assessment of TROPOMI and OMI formaldehyde observations and validation against MAX-DOAS network column measurements, Atmos Chem Phys, 21, 12561–12593, 2021.

Kim, G., Choi, Y.-S., Park, S. S., and Kim, J.: Effect of solar zenith angle on satellite cloud retrievals based on O2–O2 absorption band, Int J Remote Sens, 42, 4224–4240, 2021.

Kwon, H.-A., Park, R. J., González Abad, G., Chance, K., Kurosu, T. P., Kim, J., Smedt, I. De, Roozendael, M. Van, Peters, E., and Burrows, J.: Description of a formaldehyde retrieval algorithm for the Geostationary Environment Monitoring Spectrometer (GEMS), Atmos Meas Tech, 12, 3551–3571, 2019.

Lorente, A., Folkert Boersma, K., Yu, H., Dörner, S., Hilboll, A., Richter, A., Liu, M., Lamsal, L. N., Barkley, M., and Smedt, I. De: Structural uncertainty in air mass factor calculation for NO 2 and HCHO satellite retrievals, Atmos Meas Tech, 10, 759–782, 2017.

Xue, M., Ma, J., Tang, G., Tong, S., Hu, B., Zhang, X., Li, X. and Wang, Y.: RO x Budgets and O3 Formation during Summertime at Xianghe Suburban Site in the North China Plain, Adv. Atmos. Sci., 38(7), 1209–1222, 2021.

---

## Referee Report (RR1)

The paper describes the use and improvements of the GEMS formaldehyde retrievals. Also, it shows the results of comparison between GEMS and TROPOMI, as well as ground observations. The paper is well written and provides useful information for the scientific community. Therefore, I recommend its publication, after the minor comments below are addressed.

Specific comments
- Line 27: Is the agreement substantially higher in terms of correlation coefficient only ? or the bias is much lower in this case? Any reason why the agreement is better over Northeast Asia?

- Line 41: Could you provide the ranges for the emissions and the uncertainties?

- Line 100: How do you define clean?

- Line 110: I was wondering how sensitive the background correction is to the selection of the transport chemical model?

- Line 127: Why did you select 0.4 as a threshold for the cloud radiance fraction for the clear-sky pixels? What is the % of the pixels taken into account after masking in Fig 1?

- Line 153: There is an updated version for CEDS available. Are the emissions much different compared to the version used in the study?

 McDuffie, E. E., Smith, S. J., O'Rourke, P., Tibrewal, K., Venkataraman, C., Marais, E. A., Zheng, B., Crippa, M., Brauer, M., and Martin, R. V.: A global anthropogenic emission inventory of atmospheric pollutants from sector- and fuel-specific sources (1970–2017): an application of the Community Emissions Data System (CEDS), Earth Syst. Sci. Data, 12, 3413–3442, https://doi.org/10.5194/essd-12-3413-2020, 2020.

-Line 165: the meaning of the sentence is not very clear to me.

-Figure 5&7&8: The orientation of the labels in these plots is different from the others. The labels in the colorbal are written from top to bottom, whereas in these plots they are written from bottom to top.

- Figure 5: panel c shows the relative or the absolute difference (a-b as indicated in the caption)?

- Line 177-182: Could you explain better and elaborate about the determination of the fitting window?

- Line 256 and line 257 : Any reference for the reason for the highest HCHO? The peak for Hanoi by GEMS is not in spring but around September.

- Line 283: Why was the comparison done only for this site? Why in the comparison with TROPOMI you used the averaged values over pixels within a 20 km x 20 km grid box centered on the center of each city, while now you set a grid of 0.4 0.4 degrees?

-Line 295: is the agreement better just because GEMS and ground obs use the same info (same a priori profile)?

- Line 342 and 345: How much is the high positive bias?

- Line 358: I would give numbers for the correlation coefficient and the biases in the conclusions to summarise the main findings.

-Figure S4: The figure can be read easier if the sites are ordered by increasing longitude in the legend.

-Figures: The use of italics in the figures labels and units is not consistent in the main paper and the supplement.

---

## Author Response (AR2)

**Responses to Referee's Comments**

*We appreciate the reviewers' careful reading and insightful comments. We have carefully checked and addressed all comments and questions.*

*We have written the referee's comments in black and our responses to the comments in blue and italics. The revised sentences and paragraphs related to these comments are shown in red.*

**Referee #1:**

1. Figure 8 suggests that retrieved slant columns using irradiance as source term are smaller than differential slant columns retrieved using radiance reference. This result is contradictory with physics retrievals and with figure 8 of the original submission. This results need to be further investigated or clarified.

*Thank you for the comments. We conduct a de-striping process for the irradiance reference retrieval, which substracts latitudinally median columns of background regions (120-150°E) from each retrieved slant column, resulting in lower SCDs than the retrieved dSCDs based on radiance reference. However, the GEMS irradiance retrieval is not 100% mature and needs further improvement, especially for a de-striping process.*

2. Throughout the text and in the relevant figures, please be accurate and refer to slant column when using irradiance source term and differential slant column when using radiance reference.

*As you pointed out, we used SCDs from the irradiance retrieval and dSCDs from the radiance retrieval in the revised manuscript.*

**Referee #2:**

I have read the revision and the responses to the reviewers, and think the authors have properly addressed the reviewers' comments. I have a few additional comments.

1. The authors have considered the a priori profile difference between GEMS and ground-based retrievals, but why the a priori profile difference between GEMS and TROPOMI retrievals are not considered in the comparison of VCDs from the two satellite sensors. The a priori profile difference can lead to AMF difference and finally contribute to VCD differences.

*Thanks for the constructive comment, which we agree with. The comparison of two satellite products with identical a priori profiles is our ongoing work for future publication, and it takes some time for a long-term period of validation. In this study, we focused on comparing the operational products between GEMS and TROPOMI as they are.*

2. The caption for Fig. 5c should be "absolute difference (c) (a-b)" rather than "relative difference (c) ((a-b)/b)". The unit of the colorbar of Fig. 5c is missing.

*Thank you for pointing out the typo. We have replaced the caption of "relative difference (c) ((a-b)/b)" with "absolute difference (c) (a-b)". Also, we have added the unit of the colorbar.*

**Referee #3:**

The paper describes the use and improvements of the GEMS formaldehyde retrievals. Also, it shows the results of comparison between GEMS and TROPOMI, as well as ground observations. The paper is well written and

provides useful information for the scientific community. Therefore, I recommend its publication, after the minor comments below are addressed.

Specific comments

1. Line 27: Is the agreement substantially higher in terms of correlation coefficient only ? or the bias is much lower in this case? Any reason why the agreement is better over Northeast Asia?

*Both the correlation coefficient and NMBs are better in Northeast Asia. In Northeast Asia, there are many anthropogenic sources for HCHO, so the spectral fitting can be more accurate than clean regions. In addition, considering that GEMS is located at 127°E, viewing geometric angles in Northeast Asia are smaller than those in western areas of the GEMS domain. That could be the reason why the agreement is better over Northeast Asia.*

2. Line 41: Could you provide the ranges for the emissions and the uncertainties?

*We added the range of global biogenic VOCs emission estimates for 2000 to 2019 (Sindelarova et al., 2022).*

*Line 40: NMVOCs also play a critical role in the formation of secondary organic aerosols (DiGangi et al., 2012). They are emitted from both anthropogenic and biogenic sources (Vrekoussis et al., 2010). The latter is more significant globally but has significant uncertainty of a priori emission estimates (424–591 Tg C $yr^{-1}$) (Abbot et al., 2003; Palmer et al., 2001; Sindelarova et al., 2022).*

3. Line 100: How do you define clean?

*As shown in line 126, we defined clean pixels where cloud radiance fraction is lower than 0.4 over the background region including Pacific Ocean. We have added the term "background" into the sentence at line 100.*

*We have tested the cloud masking criterion of 0.3 and 0.2, but there are no significant improvements in the retrieved dSCDs quality. To conserve stable number of observed clen pixels sampled in radiance references, we decided to use 0.4 as an optimum criterion of the clean pixel.*

*Line 100: In spectral fitting, the measured radiances over clean background regions, referred to as radiance references, can be used instead of the solar irradiance.*

4. Line 110: I was wondering how sensitive the background correction is to the selection of the transport chemical model?

*Background HCHO concentrations are mainly derived by the oxidation of $CH_4$ over the Pacific Ocean. The model does not show dramatic differences in background columns since the simulations have no anthropogenic or biogenic emission sources over the ocean. Figure S1 from Kwon et al. (2019) shows background HCHO columns derived from the initial version of the GEMS a priori profile (Table 2). By comparing Fig. S1 (Kwon et al., 2019) with Fig. S2 (this study), background columns do not vary sensitively by the selection of the model.*

5. Line 127: Why did you select 0.4 as a threshold for the cloud radiance fraction for the clear-sky pixels? What is the % of the pixels taken into account after masking in Fig 1?

*Please refer to our response about the cloud fraction threshold above. The radiance reference pixels with cloud masking (cloud fraction < 0.4) are about ~80 % of the total radiance references.*

6. Line 153: There is an updated version for CEDS available. Are the emissions much different compared to the version used in the study?

McDuffie, E. E., Smith, S. J., O'Rourke, P., Tibrewal, K., Venkataraman, C., Marais, E. A., Zheng, B., Crippa, M., Brauer, M., and Martin, R. V.: A global anthropogenic emission inventory of atmospheric pollutants from sector- and fuel-specific sources (1970–2017): an application of the Community Emissions Data System (CEDS), Earth Syst. Sci. Data, 12, 3413–3442, https://doi.org/10.5194/essd-12-3413-2020, 2020.

*Thank you for pointing out the updates in the latest reference paper of CEDS. Our a priori profile simulations are based on the CEDS version you mentioned in the comment. We have revised the reference literature and CEDS version referred to in the paper.*

**Table 2. Summary of the input options of a priori profiles for the GEMS HCHO algorithm.**

| Version | Initial | Operational |
|---|---|---|
| Model | GEOS-Chem (v9-01-02) (Bey et al., 2001) | GEOS-Chem (v13) (Bey et al., 2001) |
| Period | 2014 | August 2020–July 2021 |
| Horizontal resolution | 2° × 2.5° | 0.25° × 0.3125° |
| Vertical layers | 47 | 47 |
| Meteorology | Modern-Era Retrospective Analysis for Research and Applications (Rienecker et al., 2011) | GEOS-FP (Goddard Earth Observing System -Forward Processing) assimilated meteorology |
| Emission inventory | **Biogenic**
- Model of Emissions of Gases and Aerosols from Nature (MEGAN) version 2.1 (Guenther et al., 2006)
**Anthropogenic**
- Database for Global Atmospheric Research version 2.0 inventory (Olivier et al., 1996)
- Mosaic fashion with the Intercontinental Chemical Transport Experiment Phase B (Zhang et al., 2009)
**Monthly biomass burning**
Global Fire Emissions Database (GFED) version 3 inventory (van der Werf et al., 2010) | **Biogenic**
- MEGAN version 2.1 (Guenther et al., 2006)
**Anthropogenic**
- Community Emissions Data System v2020-08 (McDuffie et al., 2020)
- KORUS version 5 over Asia (Woo et al., 2020)
**Monthly biomass burning**
GFED version 4 inventory (Giglio et al., 2013) |

7. Line 165: the meaning of the sentence is not very clear to me.

*We have revised the sentence for better readability.*

*Line 165: We considered the polarization sensitivity vectors of the instrument as an additional absorption cross-section, termed a pseudo absorber, in the spectral fitting.*

8. Figure 5&7&8: The orientation of the labels in these plots is different from the others. The labels in the colorbal are written from top to bottom, whereas in these plots they are written from bottom to top.

*We modified the orientation of labels in Fig. 5, 7, and 8 as written from bottom to top format.*

9. Figure 5: panel c shows the relative or the absolute difference (a-b as indicated in the caption)?

*Thank you for pointing out the typo. As mentioned in the Referee#2's comment, we revised it.*

10. Line 177-182: Could you explain better and elaborate about the determination of the fitting window?

*We added the sentence to elaborate the determination of our fitting window selection.*

*Line 180: Theoretically, retrieved dSCDs over the reference sector should be zero. We selected the fitting window of 329.3–358.6 nm for the GEMS HCHO operational retrieval based on the dSCDs closest to zero, lowest fitting residuals, and lowest fitting uncertainty (Fig. 6).*

11. Line 256 and line 257 : Any reference for the reason for the highest HCHO? The peak for Hanoi by GEMS is not in spring but around September.

*Following the statement of Baek et al. (2014), the high HCHO VCDs over Hanoi are mainly due to the emissions from anthropogenic sources. These emissions consistently elevate the HCHO concentrations throughout the year, resulting in small fluctuations from September to March. Furthermore, cloud radiance fractions continuously exceed 0.3 throughout March. Elevated low-level cloud fractions could lead to increased AMF and lower VCDs.*

12. Line 283: Why was the comparison done only for this site? Why in the comparison with TROPOMI you used the averaged values over pixels within a 20 km x 20 km grid box centered on the center of each city, while now you set a grid of 0.4 0.4 degrees?

*1. The available ground-based HCHO observation in East Asia is limited during the first year of the GEMS operation. We decided to use Xianghe MAX-DOAS and FTIR products because they cover most of these periods and provide averaging kernel and a priori profile variables, which enable precise intercomparison in satellite validation.*

*2. We acknowledge the inconsistency of the collocation criteria and have adjusted the spatial criterion for ground-based validation from a grid size of 0.4° × 0.4° to 20 km × 20 km. The general evaluation results in the updated ground validation (Figures 11, 12, S8, S10, S11) are similar to the previous comparison.*

13. Line 295: is the agreement better just because GEMS and ground obs use the same info (same a priori profile)?

*The improved agreement between GEMS and ground-based observations could be from sharing the same a priori profile and the smoothing effect of averaging kernel. We have added brief descriptions for the potential*

*discrepancies from the vertical sensitivity.*

*Line 298: In addition, different vertical sensitivities of satellite and ground-based observations could cause inevitable discrepancies in their vertical columns.*

14. Line 342 and 345: How much is the high positive bias?

*We have added the exact number of the high positive biases in the sentence.*

*Line 347: The initial algorithm caused high positive biases (10–40 %) in the slant columns from the spectral fitting, primarily due to radiance references constructed under cloudy conditions with high reflectance.*

15. Line 358: I would give numbers for the correlation coefficient and the biases in the conclusions to summarise the main findings.

*We have added the numbers for the correlation coefficient and the biases in the conclusions.*

*Line 363: We found high correlations between GEMS and TROPOMI HCHO VCDs and a good representation of seasonality with the regional characteristics of GEMS HCHO among the major cities (r=0.58–0.82), showing active emissions from biogenic and anthropogenic sources over East Asia.*

*Line 370: GEMS produced approximately 30 % lower VCDs than MAX-DOAS but showed high correlations (r=0.77) and consistent seasonality with MAX-DOAS during the year.*

*Line 372: The MAX-DOAS- and FTIR-recalculated VCDs with the a priori profiles of GEMS decreased, showing reduced NMBs (−47.4 % to −31.5 % and −38.6 % to −26.7 % for MAX-DOAS and FTIR, respectively) against GEMS.*

16. Figure S4: The figure can be read easier if the sites are ordered by increasing longitude in the legend.

*We have modified the order of legend by longitude.*

[Figure]

**Fig. S4. Regions selected for the comparison between GEMS and TROPOMI.**

17. Figures: The use of italics in the figures labels and units is not consistent in the main paper and the supplement.

*We used labels with the italic style in Fig. 4, Fig. 11, Fig. S5, and Fig. S9*

**References**

Baek, K. H., Kim, J. H., Park, R. J., Chance, K., & Kurosu, T. P. (2014). Validation of OMI HCHO data and its analysis over Asia. Science of the Total Environment, 490, 93–105.

McDuffie, E. E., Smith, S. J., O'Rourke, P., Tibrewal, K., Venkataraman, C., Marais, E. A., Zheng, B., Crippa, M., Brauer, M., and Martin, R. V: A global anthropogenic emission inventory of atmospheric pollutants from sector-and fuel-specific sources (1970–2017): an application of the Community Emissions Data System (CEDS), Earth Syst Sci Data, 12, 3413–3442, 2020.

Sindelarova, K., Markova, J., Simpson, D., Huszar, P., Karlicky, J., Darras, S., and Granier, C.: High-resolution biogenic global emission inventory for the time period 2000–2019 for air quality modelling, Earth Syst. Sci. Data, 14, 251–270, https://doi.org/10.5194/essd-14-251-2022, 2022.